# The miR-144/Hmgn2 regulatory axis orchestrates chromatin organization during erythropoiesis

Dmitry A. Kretov [1], Leighton Folkes [2], Alexandra Mora-Martin[1], Isha A. Walawalkar [1], Imrat[1], Noreen Syedah[1], Kim Vanuytsel [3,4,5], Simon Moxon [2], George J. Murphy [3,4] & Daniel Cifuentes [1,6] ✉

Differentiation of stem and progenitor cells is a highly regulated process that involves the coordinated action of multiple layers of regulation. Here we show how the post-transcriptional regulatory layer instructs the level of chromatin regulation via miR-144 and its targets to orchestrate chromatin condensation during erythropoiesis. The loss of miR-144 leads to impaired chromatin condensation during erythrocyte maturation. Among the several targets of miR-144 that influence chromatin organization, the miR-144-dependent regulation of Hmgn2 is conserved from fish to humans. Our genetic probing of the miR-144/Hmgn2 regulatory axis establish that intact miR-144 target sites in the Hmgn2 3'UTR are necessary for the proper maturation of erythrocytes in both zebrafish and human iPSC-derived erythroid cells while loss of Hmgn2 rescues in part the miR-144 null phenotype. Altogether, our results uncover miR-144 and its target Hmgn2 as the backbone of the genetic regulatory circuit that controls the terminal differentiation of erythrocytes in vertebrates.

MicroRNAs (miRNAs) are a family of small noncoding RNAs that regulate gene expression by destabilizing and repressing the translation of their target mRNAs[1]. As critical post-transcriptional regulators, miRNAs play a central role in cell differentiation and embryo development by restricting cell fate choices and regulating developmental timing[2,3]. miRNAs enact this post-transcriptional control by targeting hundreds of mRNA each[4]. However, analysis of miRNA loss-of-function mutants reveals that the bulk of expression changes come from the additional dysregulation of non-target mRNAs[1–3]. Uncovering the molecular mechanisms governing how miRNAs expand their regulatory role to additional non-target mRNAs will help to establish a more comprehensive framework of miRNA-mediated regulation of cell differentiation and development. Here, we uncover and genetically probe how the post-transcriptional regulation elicited by miR-144 instructs chromatin organization during erythropoiesis by ultimately increasing the number of transcripts affected directly or indirectly by miR-144 activity.

Erythropoiesis is a highly orchestrated and dynamic process in which multipotent hematopoietic stem and progenitor cells (HSPCs) progressively differentiate to become mature erythrocytes[5]. Two vertebrate-specific miRNAs, miR-144 and miR-451, regulate erythrocyte terminal differentiation. These miRNAs are expressed from a single primary miRNA precursor whose expression is activated by the transcription factor GATA1[6]. miR-451, which accounts for up to 60%[7] of the miRNA content in mature erythrocytes, is the only known miRNA whose processing is independent of Dicer but instead relies on the slicer activity Ago2[8–10]. miR-451 function is required for proper erythroid maturation[11,12] and mounting the oxidative stress response[13,14]. miR-144 is involved in the regulation of globin synthesis and oxidative protection of the cell[15,16]. Our recent work also demonstrated that miR-

[1]Department of Biochemistry and Cell Biology, Boston University Chobanian & Avedisian School of Medicine, Boston, MA, USA. [2]School of Biological Sciences, University of East Anglia, Norwich, UK. [3]Center for Regenerative Medicine, Boston University Chobanian & Avedisian School of Medicine, Boston, MA, USA. [4]Section of Hematology and Oncology, Department of Medicine, Boston Medical Center, Boston, MA, USA. [5]Amyloidosis Center, Boston University Chobanian and Avedisian School of Medicine, Boston, MA, USA. [6]Department of Virology, Immunology and Microbiology, Boston University Chobanian & Avedisian School of Medicine, Boston, MA, USA. ✉e-mail: dcb@bu.edu

144 establishes a negative feedback loop with Dicer that induces the global downregulation of canonical miRNAs while promoting the Dicer-independent processing of miR-451 during erythropoiesis[17].

Committed erythroid progenitors undergo a global repression of gene expression with the exception of erythroid-specific genes such as globins, proteins involved in membrane organization, and specific noncoding RNAs[18,19]. In all vertebrates, terminal differentiation of erythrocytes is accompanied by progressive chromatin condensation, which in mammals culminates with the extrusion of the nucleus[18,20,21]. These rearrangements in nuclear organization are needed to restrict cellular fate to the erythroid lineage and direct cellular machinery toward the synthesis of proteins necessary for the proper function of erythrocytes[22,23]. Perturbations of erythrocyte maturation can lead to anemia and myelodysplastic syndromes[24,25].

In our current work, we conducted a phenotype-informed search to uncover miR-144 targets that orchestrate nuclear condensation during terminal erythropoiesis. We identified a gene regulatory axis, comprising miR-144 and its direct target Hmgn2, involved in chromatin organization. Disruption of the miR-144-mediated regulation of Hmgn2 recapitulates in part the impaired erythropoiesis phenotype of miR-144 mutants. Conversely, the reduction of Hmgn2 activity restores normal erythrocyte maturation in miR-144 mutants. Overall, we show how microRNAs can expand their targeting network and amplify their regulatory potential via targeting a master regulator of chromatin organization.

## Results

### Loss of miR-144 leads to defects in nuclear condensation and genome-wide dysregulation of gene expression in erythroblasts

During terminal differentiation, erythroblasts progressively condense their chromatin, which culminates in the extrusion of the nucleus in mammals, or a reduction of the nuclear volume in all the other vertebrates, including zebrafish[19]. Our previous morphological analysis revealed that erythrocytes isolated from zebrafish mutants with a deletion in the miR-144 locus (miR-144$^{\Delta/\Delta}$) display enlarged nuclei with pronounced granular staining as compared to wild-type erythrocytes[17], a hallmark of impaired erythrocyte maturation. To determine when miR-144$^{\Delta/\Delta}$ erythrocytes first manifest nuclear defects, we conducted a time course analysis of the maturation of wild-type and miR-144$^{\Delta/\Delta}$ erythrocytes. The May–Grünwald–Giemsa (MGG) staining of erythroblasts isolated from embryos at different developmental stages (30-, 48-, and 72-hours post-fertilization (hpf)) shows that the morphology of miR-144$^{\Delta/\Delta}$ erythroblasts is indistinguishable from wild-type siblings until 48-hpf (Fig. 1A). At 72-hpf, miR-144$^{\Delta/\Delta}$ erythroblasts display an increased nucleus-to-cytoplasm area ratio (N:C) (Fig. 1A, B), typical of immature erythrocytes[12]. These results are consistent with the increased nuclear area that we found previously in erythrocytes isolated from adult fish[17] albeit at 72-hpf peripheral blood is a mix of cells produced during primitive and definitive waves of erythropoiesis. Overall, these results indicate that the deletion of miR-144 impairs erythrocyte maturation starting at 48 hpf and persisting into adulthood.

To interrogate nuclear organization in more detail, we analyzed miR-144$^{\Delta/\Delta}$ erythrocytes from 2- and 3-days post-fertilization (dpf) embryos by transmission electron microscopy (TEM). miR-144$^{\Delta/\Delta}$ erythroblasts displayed a 1.44-fold increase of light nuclear areas (euchromatin) compared to wild-type cells only at 3-dpf (Fig. 1C, D), but not at 2-dpf (Supplementary Fig. 1A, B) suggesting an impairment in heterochromatin formation during differentiation of miR-144$^{\Delta/\Delta}$ cells. Following these results, we reasoned that the prevalence of euchromatin regions could be conducive to enhanced transcription[26]. To probe this hypothesis, we quantified the levels of RNA polymerase II (RNAP II) engaged in transcription. RNAP II actively involved in transcript elongation is phosphorylated at Ser2 in its C-terminal domain[27,28]. Therefore, analysis of the phosphorylation status of

RNAP II by immunofluorescence using a specific antibody against the phosphorylated isoform is a proxy for the fraction of RNAP II engaged in transcription[28]. In concordance with our hypothesis, we observed a ~two-fold increase in the RNAP II pSer2 signal in miR-144$^{\Delta/\Delta}$ erythroblasts isolated from 3-dpf embryos as compared to their wild-type siblings (Fig. 1E, F). Altogether, these results suggest that loss of miR-144 impairs chromatin condensation of erythrocytes and leads to increased transcriptional activity.

To determine if the defect in chromatin remodeling of miR-144$^{\Delta/\Delta}$ erythroblasts is restricted to a few specific genomic loci or a genome-wide effect, we performed ATAC-seq analysis. We analyzed peripheral blood cells isolated from 2- and 3-dpf embryos and adult zebrafish. Analysis of ATAC-seq peaks indicated that there was no significant difference in peak accessibility between wild-type and miR-144$^{\Delta/\Delta}$ erythrocytes isolated from 2-dpf animals (Fig. 1H). However, we observed an increase in accessible chromatin regions in miR-144$^{\Delta/\Delta}$ erythroblasts at 3-dpf and in erythrocytes of adult fish (Fig. 1H) compared to their wild-type counterparts. This finding is in agreement with the onset of nuclear defects observed at 3-dpf (Fig. 1A) and demonstrates that the loss of miR-144 has a direct effect on chromatin organization.

To determine whether the increase in open chromatin regions leads to a global increase in gene transcription, as hinted by the RNAP II pSer2 staining, we analyzed gene expression using bulk mRNA sequencing at 3-dpf (Fig. 1G). Our analysis shows that the 21,013 genes with more than 5 scaled reads per base in any of the genotypes, 3655 genes are upregulated and 1740 genes that are downregulated more than twofold in miR-144$^{\Delta/\Delta}$ compared to wild-type erythrocytes confirming a global trend of the increase in gene expression caused by the deletion of miR-144 (Supplementary Fig. 1C).

Among the 3655 genes expressed at higher levels in the miR-144$^{\Delta/\Delta}$, only 457 contain at least one predicted miR-144 target site in their 3'UTR, according to TargetScan v6.2. In addition, we observed that 272 genes of those stabilized in miR-144$^{\Delta/\Delta}$ have a more than onefold increase in chromatin accessibility as revealed by ATAC-seq at 3-dpf. Only 38 genes with upregulated chromatin accessibility are also miR-144 targets indicating that these two regulatory programs operate largely independently. For example, we detected the expression of adult globin genes (hbaa1 and hbba1), that are not miR-144 targets, along with embryonic globins (Fig. 1G). Since adult globins are only expressed in wild-type zebrafish larvae older than ~30-days[29], these results suggest that the increase in mRNA expression in the miR-144$^{\Delta/\Delta}$ mutants is not only due to disruption of miRNA-mediated silencing and subsequent stabilization of miRNA targets but also represents a major disruption of the transcriptional program caused by alterations at the chromatin level. Moreover, in erythrocytes isolated from adult fish, we observed an even stronger correlation between an increase in chromatin accessibility and elevated gene expression in miR-144$^{\Delta/\Delta}$ (Supplementary Fig. 1D, E). This correlation coincides with the timing of the onset of chromatin condensation and highlights its important role in shaping gene expression during erythropoiesis. Moreover, these results show that the impact of miRNA-mediated regulation on gene expression goes beyond the immediate regulation of their direct targets.

### miR-144 targets several chromatin factors, including Hmgn2

We hypothesized that the impaired nuclear condensation phenotype of miR-144$^{\Delta/\Delta}$ erythrocytes may be driven at least in part by the gain of function of one or more of the miR-144 mRNA targets that are upregulated in the mutant fish. Our previous work demonstrated that miR-144 targets Dicer in erythrocytes, affecting the processing of miR-451 and all canonical miRNAs[17]. The transcriptomic analysis of peripheral blood in 2-dpf zebrafish embryos demonstrates that Dicer is among the mRNAs enriched upon loss of miR-144 (Fig. 2A). To test if Dicer stabilization and accompanied dysregulation of miRNA metabolism might contribute to the miR-144$^{\Delta/\Delta}$ phenotype, we injected mRNA

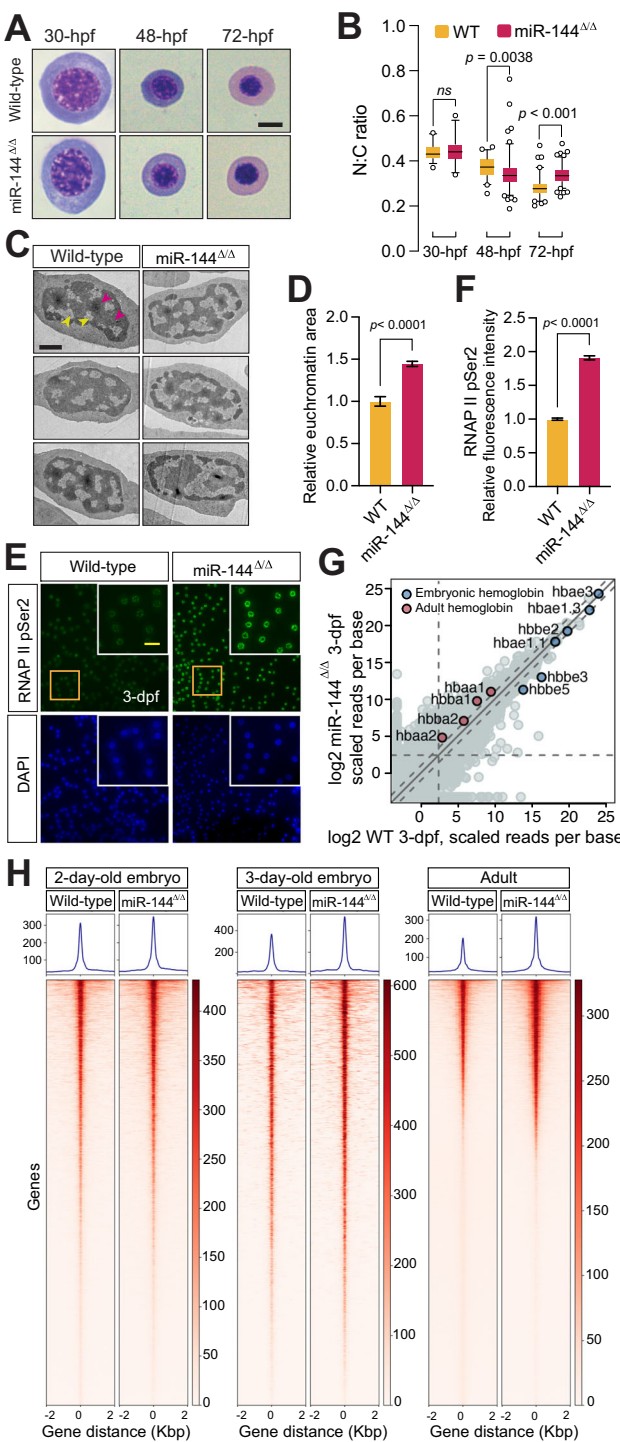

**Fig. 1 | Loss of miR-144 impairs chromatin condensation during erythropoiesis.**
**A** May–Grünwald–Giemsa staining of peripheral blood cells isolated from miR-144$^{\Delta/\Delta}$ and wild-type siblings at 30-hpf, 48-hpf, and 72-hpf. Scale bar indicates 5 µm in length. **B** Quantitative analysis of nucleus-to-cytoplasm area ratio of erythrocytes stained in (**A**). Individual cells from the pooled blood of -100 embryos, which in turn are the mixed offspring of multiple breeding pairs are analyzed in each case (wild-type: $n = 32$, $n = 51$, and $n = 81$ cells; miR-144$^{\Delta/\Delta}$: $n = 26$, $n = 113$, and $n = 108$ cells, respectively at 30, 48, and 72-hpf). *P* values from one-way ANOVA test. Boxes enclose data between the 25th and 75th percentile, with horizontal bar indicating the median. Whiskers enclose 5th to 95th percentiles. **C** Transmission Electron Microscopy of erythrocytes isolated from 3-dpf embryos (yellow arrows indicate euchromatin and magenta indicate heterochromatin). Scale bar indicates 1 µm in length. **D** Quantification of euchromatic regions of the nuclei from (**C**). $n = 31$ cells from wild-type and $n = 29$ cells from miR-144$^{\Delta/\Delta}$ mutant cells derived from a pool of bled embryos are analyzed in each case. *P* value from two-tailed unpaired *t* test equals $P < 0.0001$. Error bars represent standard error of the mean. **E** Immunofluorescent staining of erythrocytes isolated from 3-dpf embryos with anti-RNAP II Ser2 antibodies. Scale bar indicates 10 µm in length. Data representative of three independent experiments. **F** Quantification of the nuclear signal of RNAP II Ser2 from (**E**). $n = 100$ cells are analyzed from each genotype. *P* values from two-tailed unpaired *t* test equals $P < 0.0001$. Error bars represent standard error of the mean. **H** Heatmaps of ATAC-seq analysis of erythrocytes isolated from peripheral blood 2-dpf, 3-dpf, and adult miR-144$^{\Delta/\Delta}$ fish and wild-type siblings. 2-dpf and adult samples are analyzed in triplicates, and 3-dpf samples in duplicates. **G** RNA Sequencing of erythrocytes isolated from 3-dpf. Average of two biological replicates is plotted. Expression of adult and embryonic globins is shown.

144 target genes (more than twofold upregulated and more 5 scaled reads per base (Fig. 2A). Among the targets enriched for GO terms related to chromatin, we identified chromatin regulators (cdx8, napl1l4b, and hmgn2) and transcription factors (gtf2a1 and gtf2b).

To probe which of these candidates when overexpressed can recapitulate the impaired nuclear condensation of miR-144$^{\Delta/\Delta}$ erythroblasts, we injected mRNAs encoding each of the candidate genes into one-cell stage wild-type embryos. Overexpression of all new candidates increased the N:C ratio of the erythroblasts (Fig. 2B, C). These results suggest that all these candidates participate to some degree in nuclear condensation, and that the defects observed in miR-144$^{\Delta/\Delta}$ are the result of the compound gain-of-function effect of multiple genes involved in chromatin regulation. Among these candidates, High-mobility group nuclear protein 2 (Hmgn2) stands out because it: (i) has the highest expression level in erythrocytes (Fig. 2A), (ii) causes a higher N:C ratio when overexpressed (Fig. 2C), (iii) is a conserved miR-144 target from fish to human (Fig. 2E and Supplementary Fig. 2C), and (iv) it has been previously reported to play a role in the maintenance of open chromatin state[30–32]. For all these reasons and to further dissect the role of miR-144 in chromatin regulation, we decided to focus from here onward on Hmgn2.

### *hmgn2* is a conserved target of miR-144
*Pre-miR-144* can generate two mature isoforms that are variable at their 5' ends by 1 nucleotide due to the reshaping of its terminal loop[33]. According to TargetScanFish v6.2, *hmgn2* mRNA from zebrafish (*Danio rerio*) has two predicted 7mer-m8 target sites for miR-144-v1 in its 3'UTR[34] (Fig. 2E and Supplementary Fig. 2A). Moreover, the expression of *hmgn2* is stabilized in miR-144$^{\Delta/\Delta}$ mutants (Fig. 2A). To experimentally demonstrate that *hmgn2* is a direct target of miR-144, we injected a Nanoluciferase reporter fused to the zebrafish *hmgn2* 3'UTR (Nluc-*hmgn2*-3'UTR-wt) into zebrafish embryos. Only the co-injection of the reporter with miR-144 duplex repressed reporter expression (Fig. 2F). Conversely, mutation of the miR-144 target site in the 3'UTR of *hmgn2* disrupted miR-144-mediated repression of the Nanoluciferase reporter (Fig. 2F). To probe the expression domain of endogenous *hmgn2* and its regulation by miR-144, we detected the expression of *hmgn2* mRNA in 30-hpf embryos using in situ hybridization. It revealed a distinct

encoding Dicer in zebrafish embryos at the 1-cell stage (Fig. 2B) and confirmed its expression and functionality (Supplementary Fig. 2A). At 3-dpf, we isolated circulating peripheral blood cells and performed MGG staining on the blood smears. The N:C ratio of the Dicer-injected embryos was not significantly different from the non-injected siblings (Fig. 2B, C). These results suggest that Dicer is not directly involved in the nuclear condensation process, other than via its central role in miR-144 processing.

To find additional candidates, we analyzed changes in gene expression in miR-144$^{\Delta/\Delta}$ cells at 2-dpf because we reason that the morphological changes observed in 3-dpf embryo should be triggered by molecular alterations at earlier stages of development. We performed a gene ontology (GO) analysis on the group of stabilized miR-

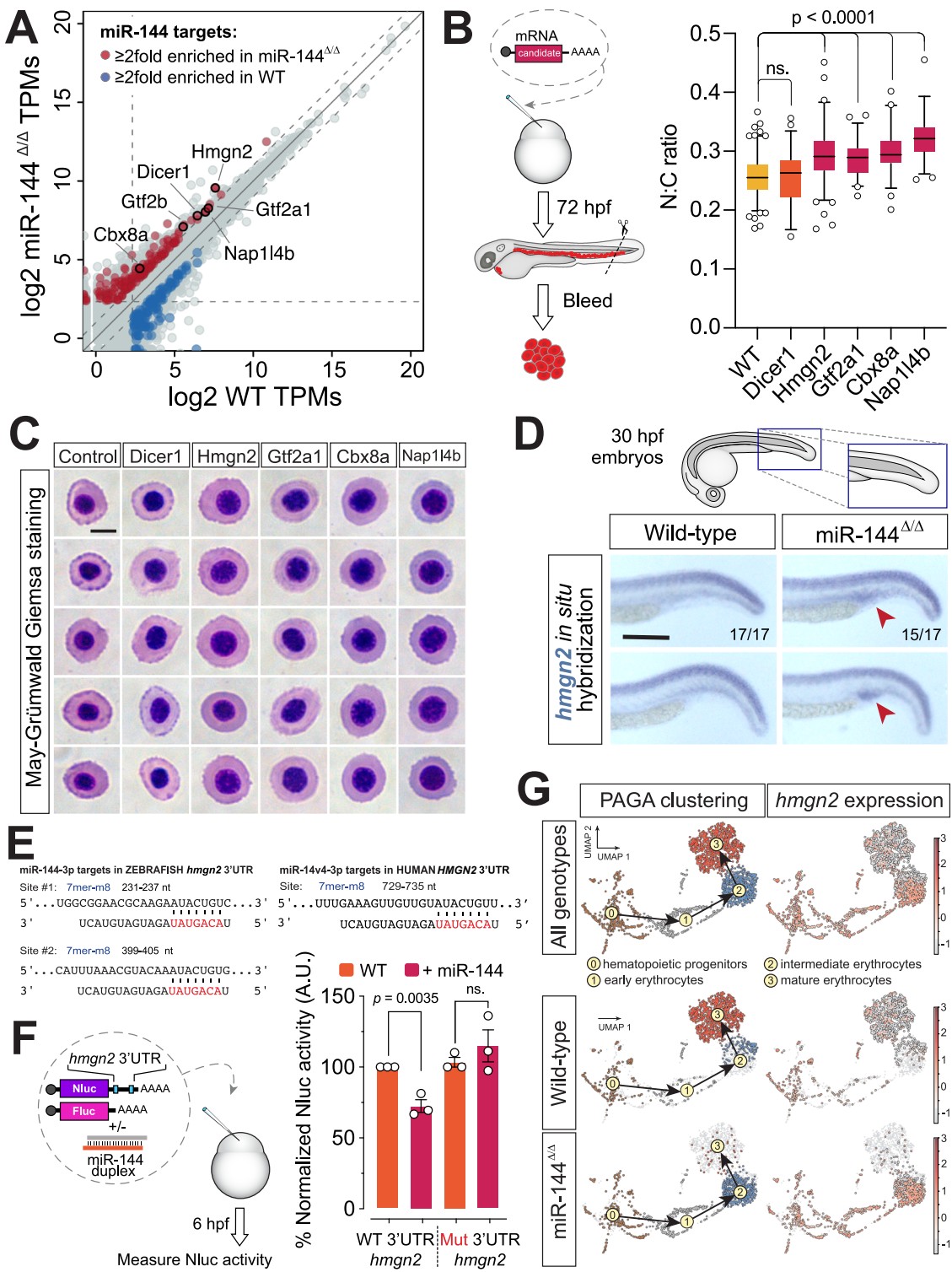

pattern of *hmgn2* expression in the notochord. In addition, expression of *hmgn2* was detected in the posterior blood island, an early hematopoietic site of the embryo, in miR-144$^{\Delta/\Delta}$ (15 out of 17 embryos) but not in wild-type siblings[35] (17 out of 17 embryos) (Fig. 2D and Supplementary Fig. 2). This observation can be explained by the stabilization of the *hmgn2* transcript in the miR-144 mutant embryos. This result demonstrates that miR-144-mediates repression of the endogenous *hmgn2* in the hematopoietic tissue and suggests that *hmgn2* downregulation occurs early in the erythrocyte maturation process.

To gain information about Hmgn2 regulation dynamics during erythropoiesis, we conducted single-cell RNA-Seq (scRNA-Seq)

analysis of the wild-type and miR-144$^{\Delta/\Delta}$ pronephros, the main hematopoietic organ of adult fish[36] (Supplementary Fig. 3A). Next, we established the developmental trajectories between clusters using scVelo[37] and used partition-based graph abstraction (PAGA)[38] to plot the arrows that indicate the inferred developmental path between clusters (Supplementary Fig. 3B, C). Focusing on the erythroid lineage, we observed that miR-144$^{\Delta/\Delta}$ erythroid cells do not reach in bulk their final differentiation state (Fig. 2G, cluster 3) and accumulate at intermediate stages when compared to wild-type cells (Fig. 2G, cluster 2). The transcriptome composition of miR-144$^{\Delta/\Delta}$ cells resembles the intermediate differentiation stages of wild-type erythroid progenitors.

**Fig. 2 | Hmgn2 is a miR-144 target whose expression is downregulated in mature erythrocytes. A** Quant-Seq data showing the transcriptomic profile of polyA-containing mRNAs in peripheral blood isolated from 2-dpf zebrafish embryos. Expression units are gene tags per million (TPMs). The average of three biological replicates is plotted. Upregulated (red) and downregulated (blue) miR-144 targets are shown. Genes involved in chromatin organization and transcription are labeled. **B** Experimental design of testing of candidate genes identified in (**A**) and quantification of the N:C ratio after May–Grünwald–Giemsa staining (**C**). Individual cells (*n* = 142 cells from wild-type, *n* = 57 from Dicer1, *n* = 113 from Hmgn2, *n* = 51 from Gtf2a1, *n* = 52 from Cbx8a, and *n* = 50 from Nap1l4b-injected embryos) derived from a pool of bled embryos at 72-hpf are analyzed. Wild-type and Hmgn2-injected data is derived from two independent pools of bled embryos. *P* values from the one-way ANOVA test with Dunnett's multiple comparisons test. Boxes enclose data between the 25th and 75th percentile, with a horizontal bar indicating the median. Whiskers enclose 5th to 95th percentiles. Cartoon is adapted with permission from ref. 17. **C** May–Grünwald–Giemsa staining of peripheral blood cells

isolated from wild-type 3-dpf embryos overexpressing different factors. Scale bar indicates 5 μm in length. **D** Whole-mount in situ hybridization of *Hmgn2* mRNA at 30-hpf of miR-144$^{\Delta/\Delta}$ fish and wild-type siblings. Blue staining indicates the presence of *hmgn2* mRNA. Red arrowhead points to the area where *hmgn2* mRNA accumulates. Scale bar indicates 200 μm in length. **E** Predicted miR-144-3p-v1 target sites in *hmgn2* 3'UTR of *Danio rerio* and *Homo sapiens*. miR-144 seed region is indicated in red. **F** Nanoluciferase reporter assays to validate miR-144 targeting *hmgn2* 3'UTR. Data represent mean ± standard error of the mean of three technical replicates. *P* values from two-tailed unpaired *t* test. Cartoon is adapted with permission from ref. 17. **G** UMAP plots illustrating the single-cell sequencing results of adult zebrafish pronephros of miR-144$^{\Delta/\Delta}$ and wild-type zebrafish. Black arrows indicate the path of developmental trajectories inferred with scVelo/PAGA. Only erythroid branch is shown. The plots are colored according to cluster identity (PAGA clustering column) or *hmgn2* expression. The expression of *hmgn2* is shown as heat map where the color scale represents the natural log of normalized read counts +1 per cell, or ln(cpm+1).

This fact indicates that the loss of miR-144 significantly disrupts the terminal differentiation process at the transcriptional level and impairs the terminal maturation of erythroblasts.

Next, we followed the expression of *hmgn2* in the developmental trajectory established using scVelo[37] from the hematopoietic progenitor cell cluster to the cluster of mature erythrocytes. We observed sustained expression of *hmgn2* in miR-144$^{\Delta/\Delta}$ cells, while the expression of *hmgn2* was reversely correlated with differentiation progression in wild-type siblings, decreasing as cells transition into mature erythrocytes (Fig. 2G). It demonstrates that loss of miR-144 leads to increased expression of *hmgn2* in erythroid progenitors and correlates with impaired differentiation progression. These results indicate that miR-144 is involved in the regulation of Hmgn2 expression in vivo at least in part through seed-mediated interactions in *hmgn2* 3'UTR.

### Disruption of miR-144-mediated regulation of *hmgn2* mimics the erythropoietic defects of the miR-144 mutant

To evaluate the importance of the miR-144/Hmgn2 regulatory axis in vivo, we genetically probed this regulatory circuit. We aimed to disrupt the direct interaction between miR-144 and Hmgn2 while preserving their expression in their cell-specific context. First, we attempted to delete the two miR-144 target sites in the endogenous *hmgn2* 3'UTR of zebrafish. However, the lack of appropriately positioned binding sites for guide RNAs in this genomic region precluded us from using a CRISPR/Cas9 system and instead drove us to apply a transgenic approach. To recapitulate the endogenous regulation of Hmgn2 by miR-144 we generated a transgenic zebrafish line expressing coding region of *hmgn2* fused to the full-length *hmgn2* 3'UTR that contains miR-144 target sites (Fig. 2E). An EYFP sequence was included in this construct to monitor the expression of the transgene and was separated from *hmgn2* by a T2A site. The 3'UTR comprising miR-144 target sites was flanked by *loxP* sites that could be used to delete by recombination the miR-144 sites of the transgene cassette and generate a transgenic line in which miR-144-mediated regulation of *hmgn2* is disrupted (Fig. 3C, D).

To ensure that the *hmgn2*-EYFP transgene has the same spatial and temporal expression as miR-144, we cloned and used the miR-144/451 promoter to drive the expression of the transgene (Fig. 3A). We defined the miR-144/451 promoter as the 5.5-kb region upstream of the start of transcription of the miR-144/451 cluster that encompasses two GATA1 binding sites that were characterized previously[6]. First, we confirmed that this promoter drives the expression of a fluorescence reporter gene (EYFP) specifically in the erythrocytes. When we crossed the *miR-144/451::EYFP* line to the *gata1::dsRed* line we detected a perfect overlap in the expression of EYFP and dsRed (Fig. 3A). We used this validated promoter to drive the expression of the aforementioned transgenic construct *miR-144/451::hmgn2-T2A-EYFP-3'UTR-WT*. We observed that the expression of EYFP was barely detectable at 24-hpf in

the wild-type background (Fig. 3E). This could be explained by repressive elements present in its 3'UTR, such as miR-144 target sites. To test this hypothesis, we injected mRNA encoding Cre recombinase in part of the F1 generation offspring of *miR-144/451::hmgn2-T2A-EYFP-3'UTR-WT* to generate the *miR-144/451:: hmgn2-T2A-EYFP-3'UTR-MUT* line that carries a truncated 3'UTR without the miR-144 target sites. Cre-mediated recombination of the transgene is highly efficient, as shown by PCR of the transgene 3'UTR, leading to 100% efficiency of *hmgn2* 3'UTR deletion (Fig. 3D). This experimental design allows us to compare two zebrafish lines with different 3'UTRs avoiding variability due to positional effects caused by independent integration of the transgene cassette at different genomic regions. Live imaging of transgenic embryos at 30-hpf revealed that *miR-144/451::hmgn2-T2A-EYFP-3'UTR-MUT* embryos express significantly higher levels of EYFP in the posterior blood island and caudal vein when compared to transgenic embryos carrying the wild-type *hmgn2* 3'UTR (Fig. 3E). These results indicate that *hmgn2* expression is regulated in vivo in the erythrocytes by *cis* elements contained in its 3'UTR, which include the miR-144 sites.

To gain quantitative information about Hmgn2 transgene expression, we performed RT-qPCR analysis using a pair of oligonucleotides specific for the EYFP embedded in the *hmgn2-T2A-EYFP* transgenic cassette. We compared the expression of *hmgn2-T2A-EYFP* with and without the *hmgn2* 3'UTR. We observed that Cre-mediated deletion of the 3'UTR leads to over fourfold upregulation of transgene expression (Fig. 3F). This observation indicates that miR-144 is a strong repressor of *hmgn2* expression in erythrocytes and validates the observation obtained by fluorescent microscopy (Fig. 3E). Nevertheless, we cannot exclude that additional factors may also contribute to *hmgn2* mRNA downregulation.

We decided to evaluate the consequences of Hmgn2 upregulation on erythrocyte morphology. We isolated peripheral blood cells from the *miR-144/451::hmgn2-EYFP-3'UTR-MUT* in a wild-type background, which expresses miR-144. We observed that erythrocytes overexpressing *hmgn2* displayed a higher N:C ratio as compared to erythrocytes from wild-type zebrafish (Fig. 3G, H). Altogether, these results provide a genetical validation that post-transcriptional regulation of *hmgn2* expression is necessary for proper erythrocyte maturation and that the disruption of this regulation can mimic the loss of miR-144 phenotype (Fig. 3I).

### Loss of function of *hmgn2* rescues the erythropoietic defects of miR-144 mutant

To counterbalance the previous experiments, we set out to rescue the miR-144 mutant genetically. We hypothesized that reducing the dosage of Hmgn2 in the miR-144$^{\Delta\Delta}$ background would compensate for the lack of post-transcriptional downregulation of Hmgn2 to an extent that may restore normal erythrocyte maturation. To this end,

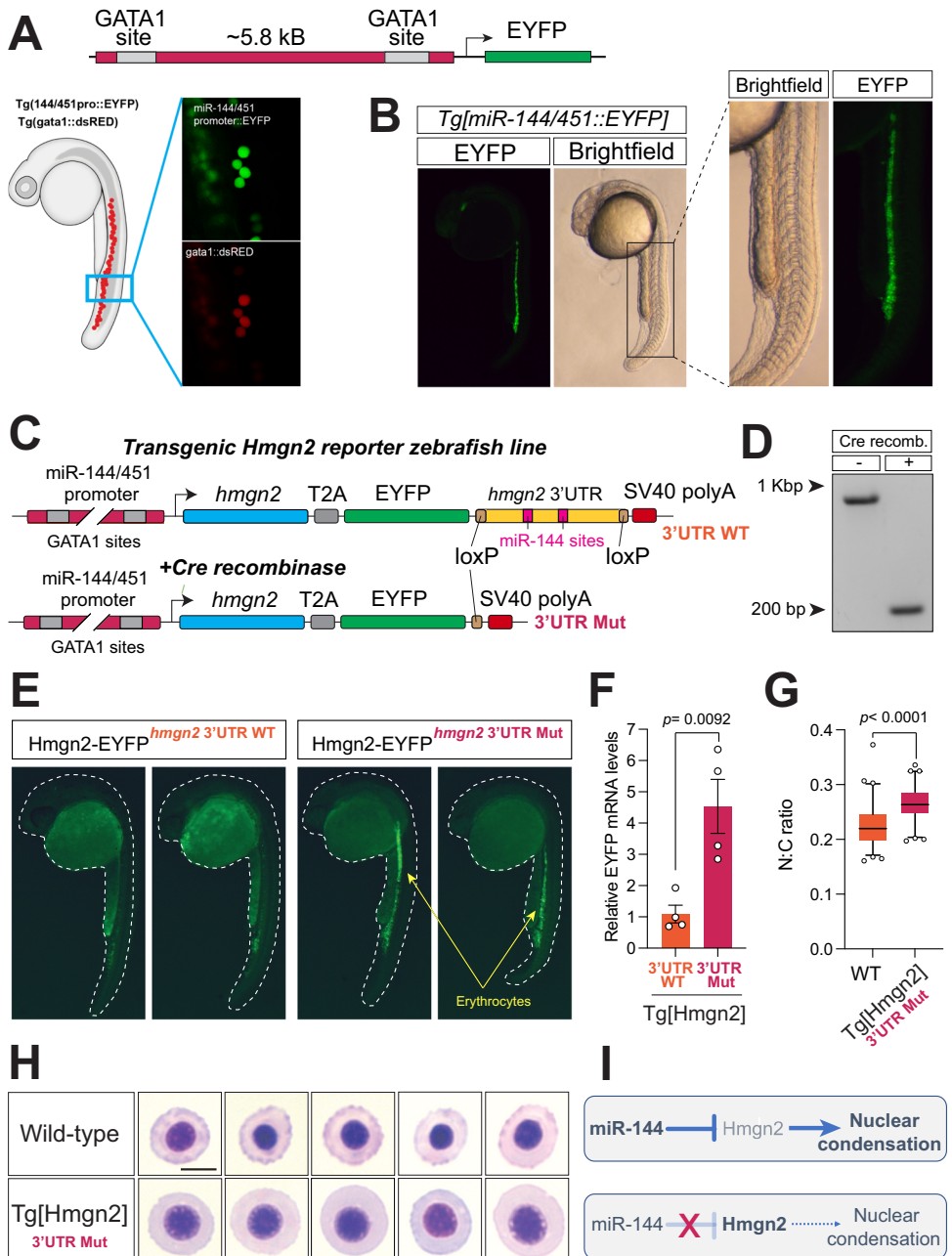

**Fig. 3 | Disruption of miR-144-mediated repression of Hmgn2 impairs erythropoiesis. A** A schematic representation of a transgene expressing EYFP from miR-144/451 promoter and its validation by colocalization with dsRed that is expressed from gata1 promoter in double transgenic fish. Photos are taken at 24-hpf. **B** miR-144/451 promoter drives the expression of EYFP in the posterior blood island and caudal vein. **C** A schematic representation describing a strategy of generating a transgenic line to analyze the expression of Hmgn2 in zebrafish. **D** PCR validation of efficient 3'UTR deletion by Cre-mediated recombination. Data representative of two independent recombination experiments. **E** EYFP expression in 30-hpf old embryos with (right) and without (left) *Cre* mRNA injection. Data representative of two independent recombination experiments. **F** Real-time quantitative PCR of Hmgn2-EYFP transgene in control and injected with *Cre* mRNA embryos at 24-hpf. Expression is normalized to *GAPDH* mRNA levels. Data represent mean ± standard error of the mean of four biological replicates. *P* value from two-tailed unpaired *t* test equals *P* = 0.00092. **G** Quantitative analysis of the N:C ratio of erythrocytes stained in (**H**). Individual cells (*n* = 68 cells from wild-type and *n* = 62 cells from TgHmgn2) derived from a pool of bled embryos are analyzed in each case. *P* value from two-tailed unpaired *t* test equals *P* < 0.0001. Boxes enclose data between the 25th and 75th percentile, with horizontal bar indicating the median. Whiskers enclose 5th to 95th percentiles. **H** May–Grünwald–Giemsa staining of peripheral blood cells isolated from the transgenic embryos with mutant *hmgn2* 3'UTR and wild-type embryos at 3-dpf. Scale bar indicates 5 μm in length. **I** A schematic describing miR-144/Hmgn2 regulatory axis. In wild-type embryos, the expression of miR-144 actively represses Hmgn2 expression. This reduction on Hmng2 levels allows nuclear condensation to proceed normally. On the other hand, expression of Hmgn2 in the miR-144 mutant increases, which prevents normal nuclear condensation.

we generated a 75-nucleotide deletion spanning intron 1 and exon 2 of *hmgn2* using CRISPR/Cas9 (Fig. 4A). According to high-throughput sequencing data, this deletion still allows for the in-frame expression of *hmgn2* skipping exon 2, which deletes 12 out of 75 amino acids of

*hmgn2* (Supplementary Fig. 4A). This Hmgn2 variant, albeit does not impair gross morphological development (Fig. 4B), appears to be non-functional because erythrocytes from homozygous maternal- and-zygotic *hmgn2* mutant (*hmgn2*$^{\Delta/\Delta}$) embryos show premature

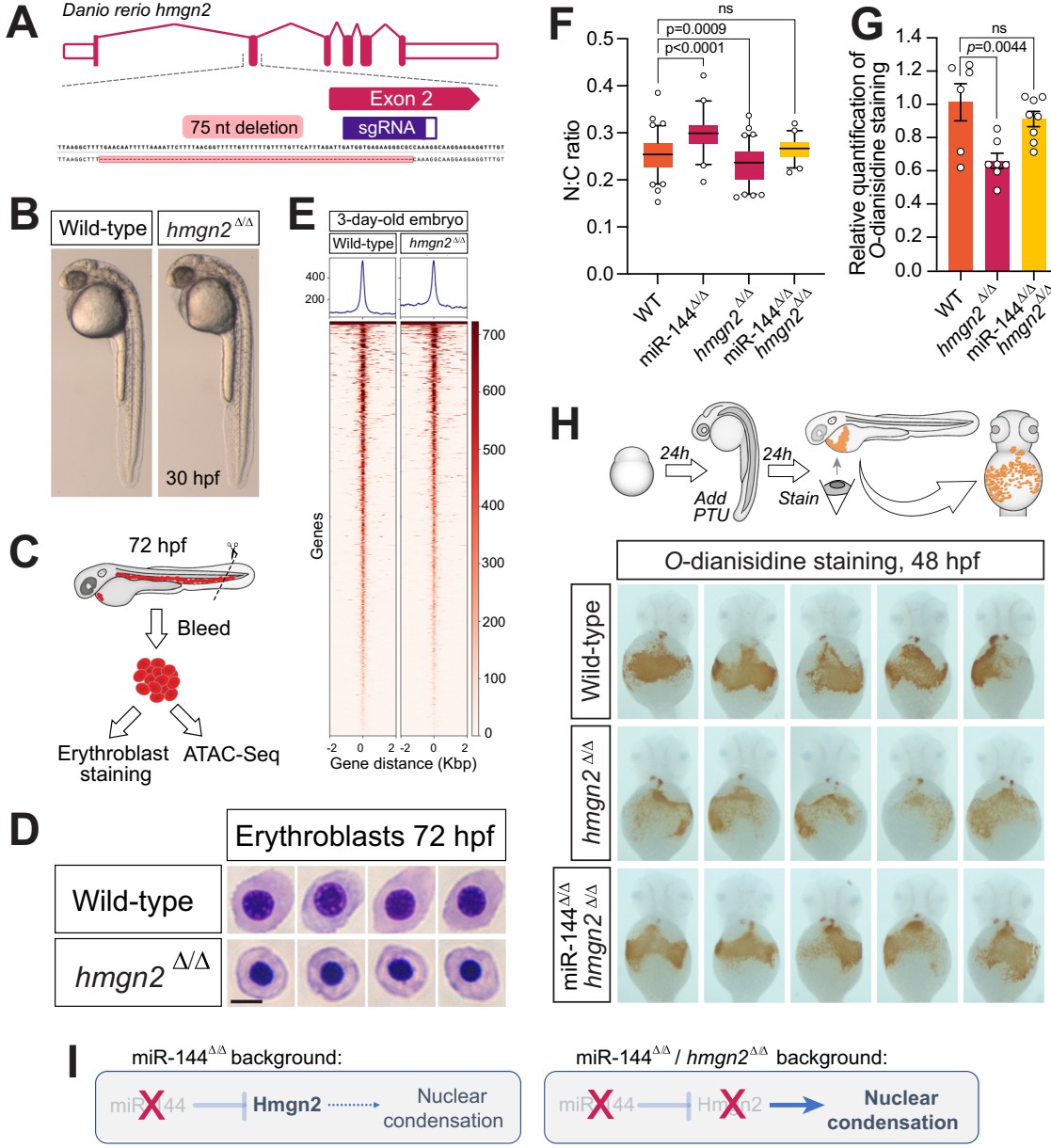

**Fig. 4 | Reduction of Hmgn2 activity rescues the loss of miR-144 in erythropoiesis. A** Schematic representation of *hmgn2* gene from *Danio rerio* and CRISPR/Cas9 induced deletion. **B** Maternal-zygotic Hmgn2 (*hmgn2*^Δ/Δ^) mutants at 30-hpf. **C** Cartoon describing the collection of peripheral blood from embryos at 72-hpf for ATAC-Seq and cellular analysis. Cartoon is adapted with permission from ref. 17. **D** May–Grünwald–Giemsa staining of peripheral blood cells isolated from *hmgn2*^Δ/Δ^ and wild-type siblings at 72-hpf. Scale bar indicates 5 μm in length. Data representative of two independent bleeding and staining experiments. **E** Heatmaps display variations in chromatin accessibility, determined through ATAC-Seq analysis of three separate samples obtained from erythrocytes isolated from peripheral blood at 72 h post-fertilization (hpf) in both *hmgn2*^Δ/Δ^ and wild-type sibling embryos. **F** Quantitative analysis of the nucleocytoplasmic ratio of erythrocytes isolated from miR-144^Δ/Δ^, miR-144^Δ/Δ^/*hmgn2*^Δ/Δ^ and wild-type siblings stained with May–Grünwald–Giemsa to rescue miR-144^Δ/Δ^ phenotype. Individual cells (*n* = 89 cells from wild-type, *n* = 41 from miR-144^Δ/Δ^, *n* = 85 from *hmgn2*^Δ/Δ^, and *n* = 45 from 144^Δ/Δ^/*hmgn2*^Δ/Δ^) from pools of bled embryos are analyzed in each case. Wild-type

and *hmgn2*^Δ/Δ^ data are derived from two independent pools of bled embryos. *P* values from one-way ANOVA with Dunnett's multiple comparisons test. Boxes enclose data between the 25th and 75th percentile, with a horizontal bar indicating the median. Whiskers enclose 5th to 95th percentiles. **G** Relative quantification of the area stained with *O*-dianisidine in the embryos shown in (**H**). Data represent mean ± standard error of the mean. *P* values from one-way ANOVA with Dunnett's multiple comparisons test. **H** *O*-dianisidine staining of 2-dpf embryos to reveal hemoglobinized cells. Embryos are pre-treated with PTU to induce mild oxidative stress to exacerbate any defect in erythropoiesis. *n* = 6 of wild-type, *n* = 7 of *hmgn2*^Δ/Δ^, and *n* = 8 of miR-144^Δ/Δ^/*hmgn2*^Δ/Δ^ individual embryos are analyzed. **I** Schematic of genetic probing of miR-144/*hmgn2* regulatory axis. In miR-144 mutant embryos, the absence of miR-144-mediated repression drives Hmgn2 overexpression, which in turn leads to reduced chromatin compaction. In the double miR-144^Δ/Δ^/*hmgn2*^Δ/Δ^ embryos, the absence of Hmgn2 activity rescues the effect of the loss of miR-144 in nuclear condensation.

nuclear condensation compared to wild-type siblings (Fig. 4C, D). These results suggest that our *hmgn2*^Δ/Δ^ is at least a partial loss-of-function mutant and indicate that full-length Hmgn2 is necessary to maintain proper nuclear organization during erythropoiesis. When we performed ATAC-Seq on erythrocytes isolated from *hmgn2*^ΔΔ^ is

embryos at 72-hpf, we observed a slight decrease in open chromatin regions which opposes the phenotype observed in the miR-144 mutant (Figs. 4E and 1H). Altogether these results confirm the role of Hmgn2 as a factor necessary to maintain an open chromatin state. Next, we crossed *hmgn2*^Δ/Δ^ mutant fish with miR-144^Δ/Δ^ to generate

the double-mutant zebrafish line miR-144$^{\Delta/\Delta}$/hmgn2$^{\Delta/\Delta}$. When we probed erythrocyte development in 72-hpf embryos, we observed that the N:C ratio of erythrocytes of this double mutant is similar to the ratio from wild-type siblings (Fig. 4F), indicating that reduced Hmgn2 activity rescues the effect of the loss miR-144 and suggesting that Hmgn2 is an important player in driving the condensation defects of the miR-144 mutant. However, if Hmgn2 were the only factor driving the N:C ratio, we would expect that the N:C ratio of the double mutant would mimic that of single hmgn2 mutant due to the inability to maintain open chromatin state even before miR-144 is expressed. Since this is not the case, these results suggest that other factors that are upregulated in the miR-144 mutant (Fig. 2A) regulate the N:C ratio of the cells. Unexpectedly, the rescue of nuclear condensation defects of the double mutant (Fig. 4F) does not translate into changes in chromatin accessibility analyzed by ATAC-seq (Supplementary Fig. 4B). We hypothesize that in the double mutant miR-144$^{\Delta/\Delta}$/hmgn2$^{\Delta/\Delta}$ there are still other chromatin factors that are expressed above normal levels (Fig. 2A, B) that may contribute to this pervasive open chromatin status detected by ATAC-Seq. These results suggest that the morphological rescue (Fig. 4F) is driven by additional mechanisms triggered by the loss of miR-144.

Mild oxidative stress can reveal underlying hematopoietic defects that remain otherwise hidden in non-sensitized backgrounds[14,17]. Here we use phenylthiourea (PTU) treatment to induce mild oxidative stress[14,17,39,40]. PTU is known to in part inhibit dopamine b-hydroxylase and tyrosinase. This blockade results in the increase of cytotoxic intermediate products like quinones, semiquinones and reactive oxygen species[39]. O-dianisidine staining after PTU treatment revealed that hmgn2$^{\Delta/\Delta}$ embryos display mild anemia under oxidative stress conditions at 48 hpf compared to wild-type embryos (Fig. 4G, H). This defect is compensated for by the additional loss of miR-144 (Fig. 4E), probably through the stabilization of other chromatin factors described before and is in agreement with the rescue of the erythrocyte N:C ratio (Fig. 4F). Altogether, these results suggest that reducing levels of wild-type Hmgn2 prevents the formation of the phenotype induced by the loss of miR-144 and re-establishes normal erythrocyte maturation (Fig. 4I).

## miR-144/HMGN2 regulatory axis facilitates differentiation of human erythroid progenitor cells

To test the relevance of the miR-144/HMGN2 axis during human erythropoiesis, we took advantage of a cell derived from human induced pluripotent stem cells (iPSCs) that can recapitulate erythroid differentiation in vitro. These iPSC-derived erythroid progenitors represent cells captured at a proerythroblast stage that can undergo further erythroid differentiation according to the previously established protocol[41] (Fig. 5A).

To test the effect of miR-144 on HMGN2 expression in human cells and its importance for erythroid differentiation, we applied a similar strategy as we used in zebrafish. We uncoupled HMGN2 from miR-144-mediated regulation by expressing HMGN2 from a lentivirus construct encoding either the wild-type human Hmgn2 3'UTR (hsHMGN2-EYFP-3'UTR-WT) or the 3'UTR with a small deletion removing miR-144 target site (hsHMGN2-EYFP-3'UTR-MUT) (Fig. 5B). As a control, we infected erythroid cells with a lentivirus expressing EYFP alone.

First, we determined that iPSC-derived erythroid progenitors express miR-144 and observed a significant upregulation of its expression, as well as of its cluster neighbor miR-451, during differentiation (Fig. 5C).

Next, we probed how miR-144 may modulate our lentiviral constructs. Cells expressing the construct with wild-type HMGN2 3'UTR display a modest increase in HMGN2-EYFP expression (twofold over control), while cells expressing transgene mRNA with mutant HMGN2 3'UTR expressed HMGN2 15-fold higher over the control (Fig. 5D, E). We confirmed that the expression of EGFP or HMGN2 did not interfere with

the levels of miR-144 in cells expressing lentiviral constructs (Fig. 5F), suggesting that the difference in the expression of HMGN2-EYFP is not due to different levels of miR-144 in these cells, but would rather be caused by the deleted target site in its 3'UTR. Overall, these results indicate that human HMGN2 is also a bona fide target of miR-144.

Next, we exposed all three engineered lines to maturation media to induce further erythropoietic differentiation. We evaluated expression of several markers at different time points to determine the differentiation progress of each cell line. We measured the expression of CD235 (Glycophorin A), a hallmark of erythroid-specific lineage[42]. We detected the expression of CD235 by flow cytometry in all samples at the end of the differentiation regime (day 7). The number of cells expressing CD235 were inversely correlated with HMGN2 expression (Fig. 5G and Supplementary Fig. S5), with hsHMGN2-EYFP-3'UTR-MUT showing the largest reduction in CD235$^+$ cells. We also quantified the expression of globins at 2 and 7 days after the start of differentiation (Fig. 5H). We observed that the expression of globins HBA ($P < 0.0001$), HBG ($P = 0.0417$), and HBE ($P = 0.0015$), but not HBB ($P = 0.3536$), was impaired between the different genotypes. Altogether, these results demonstrate that sustained opening of the chromatin impairs human erythrocyte maturation and suggest that the miR-144/HMGN2 regulatory axis is functional in humans and required for proper erythropoiesis. Future loss-of-function studies will address if miR-144 has additional targets that complement HMGN2 function in humans.

## Discussion

Here we report the unique finding that miR-144 single-handedly regulates multiple chromatin factors to force erythrocyte progenitors into the differentiation pathway. In particular, we identified and genetically probed how a miR-144/Hmgn2 regulatory axis governs nuclear condensation during erythropoiesis. Hmgn2 is a chromatin regulator that binds to nucleosomes and maintains chromatin in a transcriptionally active state[30,31,43,44]. Our data show that sustained expression of hmgn2 in the erythrocyte lineage after deletion of miR-144 target sites in its 3'UTR mimics the miR-144 mutant phenotype in zebrafish erythroblasts and human iPSC-derived erythroid cells. Conversely, we demonstrate that disruption of hmgn2 activity partially rescues the nuclear defects caused by the loss of miR-144. These results clearly intersect with previous findings showing that hmgn2 preferentially binds to chromatin regulatory sites, including the binding to mononucleosomes containing the adult globin gene cluster[45] and its role as a transcriptional activator[46]. Our data expands the impact of miR-144 activity well beyond its seed-matching target mRNAs and illustrate the broad impact of miRNA activity in shaping global gene expression. Ultimately, the significance of these results is that they challenge the current dogma in the field that perceives chromatin-modulating microRNAs as gatekeepers of cell identity and inhibitors of erythrocyte differentiation[7,47,48]. Instead, we present compelling evidence that miR-144 plays a pivotal role in enhancing the process of erythrocyte maturation from fish to humans. This paradigm shift not only enriches our understanding of miRNA-mediated regulatory mechanisms during development but also will open therapeutic opportunities to treat erythropoiesis-related disorders.

While the downregulation of multiple miRNAs has been implicated in the post-transcriptional regulation of erythrocyte maturation, here we present an opposite example whereby the activity of the miRNA is necessary to proceed through the erythrocyte maturation path. For instance, decreasing miR-191 levels are necessary to upregulate the expression of Riok3 and Mxi1, which in turn antagonize the activity of histone acetyltransferase Gcn5 and facilitate chromatin condensation[7]. Similarly, miR-181, miR-30a, miR-34a, and miR-9 suppress chromatin condensation, and their downregulation is required for normal erythrocyte maturation and nuclear extrusion[47,48]. Contrary to these known examples, the expression of miR-144 during erythropoiesis is necessary to induce nuclear condensation and subdue

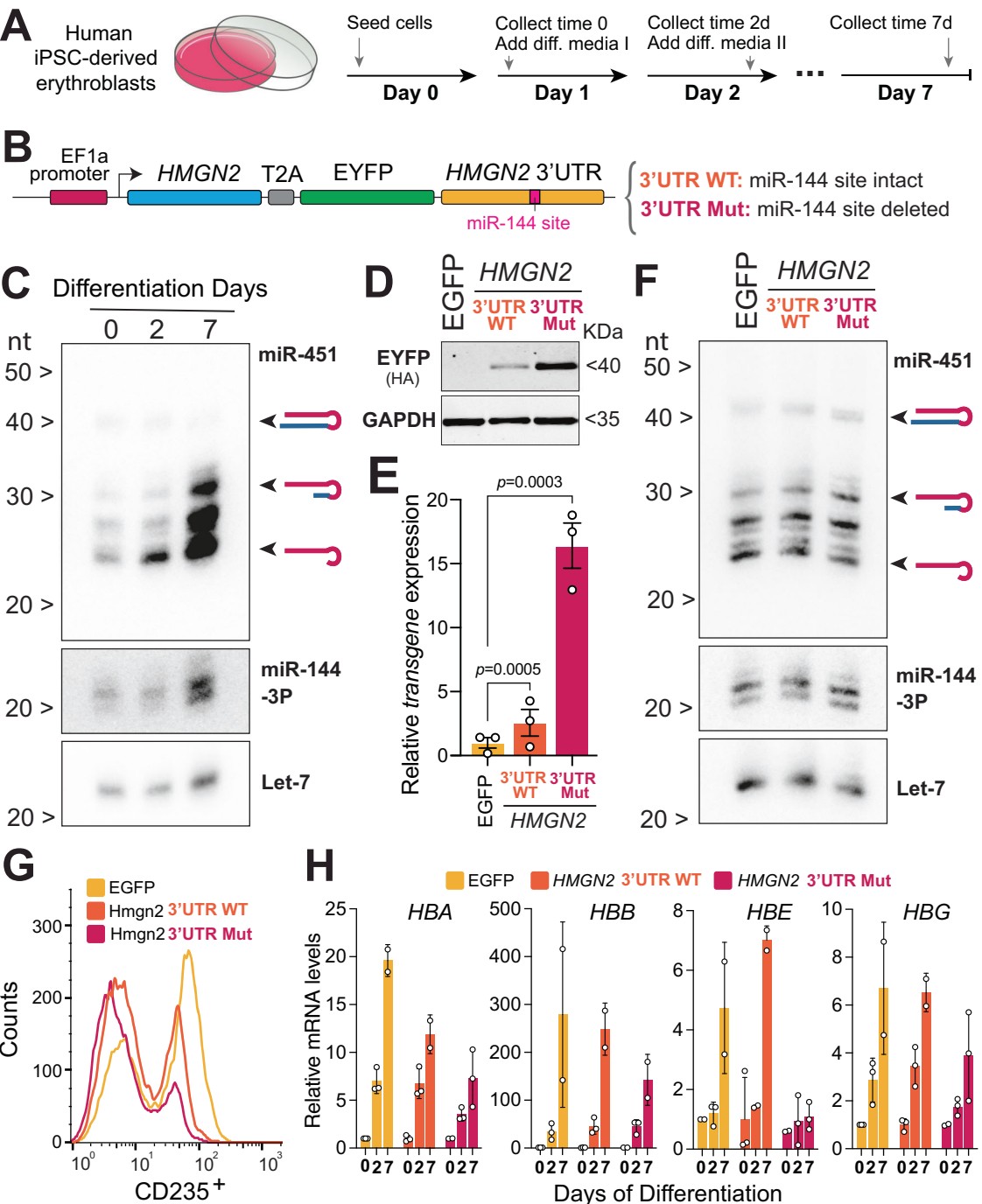

**Fig. 5 | Probing the HMGN2/miR-144 regulatory axis in human erythroid progenitor cells. A** Experimental protocol to further induce the differentiation of human iPSC-derived erythroblast cells. Cartoon is adapted with permission from ref. 17. **B** A schematic representation describing a strategy of generating cell lines to analyze the expression of Hmgn2 in human iPSC-derived erythroblast cells. **C** Northern blot analysis to detect miR-451, miR-144, and let-7 during the time course of cell differentiation. Blot representative of three biological replicates. To see the relative position of miR-144 and let-7 to the molecular weight markers please refer to the uncropped gel in Source Data file. **D** Western blot to detect accumulation of EYFP which is expressed from the transgenic construct either with wild-type 3'UTR or with mutant miR-144 target site at the time point 0. Blot representative of three biological replicates. **E** Real-time quantitative PCR of Hmgn2-EYFP transgene in at the time point 0. Expression is normalized on *GAPDH* mRNA. Data represent mean ± standard error of the mean of three biological

replicates. *P* values from one-way ANOVA. **F** Northern blot analysis to validate that overexpression of transgenic construct does not affect miR-451 and miR-144 levels at the time point 0. Blot representative of three biological replicates. To see the relative position of miR-144 and let-7 to the molecular weight markers please refer to the uncropped gel in Source Data file. **G** Flow cytometry analysis (FACS) of erythroid cells at the 7-day of differentiation to quantify the expression levels of CD235. **H** Real-time quantitative PCR of globin A (HBA), globin B (HBB), globin E (HBE), globin E (HBG) at indicated collection time points. Expression is normalized on *GAPDH* mRNA. Data derived from $n = 2$ or $n = 3$ biological replicates. Bars represent mean ± standard error of the mean. We fitted the data to a mixed model with a Geisser–Greenhouse correction, to reject the null hypothesis that all genotypes have the same population means of globin expression for HBA ($P < 0.0001$), HBG ($P = 0.0417$), and HBE ($P = 0.0015$), but not HBB ($P = 0.3536$).

transcription. This previously unappreciated role of miR-144 in chromatin regulation neatly complements the function of miR-144 that we recently described as a master repressor of canonical microRNAs via its direct targeting of *dicer1* mRNA in erythrocytes[17]. Altogether, it presents a picture whereby the combined action of multiple miRNAs regulates a cell's entry into the erythrocyte differentiation pathway. It is the role of miR-144 to simultaneously downregulate these gatekeeper miRNAs by targeting *dicer1*[17] and to collaborate with other parallel mechanisms[49] to repress *hmgn2*, thereby inducing erythrocyte maturation via chromatin remodeling.

Erythrocyte maturation is heavily regulated at the chromatin level as nuclear condensation is an essential part of the process. In mammals, this process culminates with the extrusion of the nucleus[50]. A global decrease in histone acetylation (such as H3K9Ac, H4K5Ac, H4K12Ac, and H4K8Ac) and increased methylation (H3K79 and H4K20) have been reported to be important for normal chromatin condensation[51–53]. In addition to epigenetic changes, the release of histones mediated by caspase-3 cleavage and ubiquitin-dependent degradation of nuclear lamin B[54,55] is also required for proper nuclear compaction during erythropoiesis. In the current work, we identify another regulatory mechanism that contributes to chromatin condensation during erythropoiesis. We demonstrate that repression of *hmgn2* and other chromatin factors allows chromatin condensation to proceed. High-mobility group nuclear proteins are a family of highly abundant non-histone chromatin binding proteins that modulate chromatin organization[32,56]. Among them, proteins of the HMGN family specifically remodel chromatin by antagonizing the binding of linker histone H1 to nucleosomes near chromatin regulatory sites[31,32]. From this vantage point, HMGN proteins play an important role in development. Their expression is high in mouse embryonic stem cells (ESCs)[57] and iPSCs[31], but gradually decays during tissue differentiation[58]. This phenomenon has been described for the development of multiple tissues, including the eye[59,60], hair follicle[61] and during myogenesis[62], chondrocyte differentiation[63], oligodendrocyte lineage specification[57], and erythropoiesis[64]. The expression dynamics of HMGN proteins suggest that their collective activity is necessary to maintain stem cell identity[30] but it must be dampened to proceed with cell differentiation. While changes in the chromatin structure of the HMGN genes during erythrocyte differentiation reduce in part the expression of these genes[49], here we present a parallel mechanism that downregulates *hmgn2* expression post-transcriptionally. Future work will establish if the regulation of *hmgn2* expands to other members of the high-mobility group nuclear protein family regulated by miR-144 or other miRNAs as a prevalent regulatory mode during the differentiation of other tissues.

## Methods

### Zebrafish strains
Zebrafish strains were bred, handled, and maintained according to the standard laboratory conditions under IACUC protocol PROTO201800373 at Boston University. Experiments were performed in hybrid wild-type strain crosses obtained from AB/TU and TL/NIHGRI breeders. The sex of zebrafish embryos was not considered because sex determination occurs at later time points of development. For the analysis of peripheral blood of adult zebrafish, blood was collected from both males and females. For single-cell sequencing experiment of adult zebrafish, pronephros were collected from one male fish for each genotype.

### Generation of mutant zebrafish lines using CRISPR/Cas9
miR-144 mutant line (miR-144$^{\Delta/\Delta}$) was generated and described previously in our laboratory[17]. To generate Hmgn2$^{\Delta/\Delta}$, we designed sgRNA targeting exon 2 of *hmgn2* gene was designed using CRISPRscan[65] yielding the following target sequence: 5′-GATGGTGAGAAGGGC GCCAA(AGG)-3′; where the Cas9 PAM sequence (NGG) is between

parentheses. sgRNA templates were generated by annealing and polymerase-mediated extension of a forward oligo containing the T7 promoter sequence, the 20 nt sgRNA target sequence (without the PAM sequence) and a 15 nt sequence complementary to the reverse oligo containing the invariable Cas9-binding scaffold. PCR reactions with Q5 high-fidelity polymerase (NEB) were carried out as follows: 1 cycle 95 °C for 3 min; 35 cycles (95 °C for 3 min, 55 °C for 30 s, 72 °C for 20 s); 1 cycle at 72 °C for 5 min. Reactions were purified with a PCR purification kit (NEB). Approximately 120–150 ng of DNA was used as a template for a T7 in vitro transcription (IVT) reaction with the AmpliScribe-T7-Flash transcription kit (Epicenter ASF3507). IVT sgRNA products were purified Monarch PCR and DNA Cleanup Kit (NEB) and quantified. Zebrafish embryos were injected at one-cell stage with *Cas9* mRNA (100 pg) together with 30 pg of sgRNA to generate *hmgn2* mutant. miR-144$^{\Delta/\Delta}$/Hmgn2$^{\Delta/\Delta}$ mutant was generated by crossing miR-144$^{\Delta/\Delta}$ to Hmgn2$^{\Delta/\Delta}$ line, followed by genotyping and subsequent incrossing until obtaining the miR-144$^{\Delta/\Delta}$/Hmgn2$^{\Delta/\Delta}$ fish line.

### Generation of transgenic zebrafish lines
To generate transgenic zebrafish line expressing a fluorescent EYFP reporter in erythrocytes we first cloned the miR-144/451 promoter, the 5.4 kbp region located upstream of miR-144/451 locus. We cloned pDEST-miR-144/451-EYFP plasmid from p5E-miR-144/451 promoter, pME-EYFP and p3E-polyA plasmids using Gateway LR Clonase II Enzyme Mix (Invitrogen). We generated *miR-144/451::EYFP* transgenic line using Tol2 transposon system[66]. To validate miR-144/451 promoter, *miR-144/451::EYFP* line was crossed *gata1::dsRed* line[67]. Then we generated pDEST-miR-144/451-*dre*Hmgn2-EYFP-*dre*Hmgn2–3′UTR-WT plasmid from p5E-miR-144/451 promoter, pME-*dre*Hmgn2-EYFP and p3E-*dre*Hmgn2–3′UTR-WT-polyA using Gateway LR Clonase II Enzyme Mix (Invitrogen). We included two loxP site surrounding to *dre*Hmgn2–3′UTR-WT. We generated miR-144/451-*dre*Hmgn2-EYFP-*dre*Hmgn2–3′UTR-WT line using Tol2 transposon system. In order to generate *miR-144/451::Hmgn2-EYFP-3′UTR-MUT* line, lacking most of it 3′UTR including miR-144 target sites, we injected mRNA encoding Cre recombinase (1 nL of 5 ng/µL) and validated 3′UTR removal by PCR.

### Transmission electron microscopy
The cells were fixed in 2.5% glutaraldehyde, 3% paraformaldehyde with 5% sucrose in 0.1 M sodium cacodylate buffer (pH 7.4), pelleted, and post fixed in 1% OsO$_4$ in veronal-acetate buffer. The cells were stained en block overnight with 0.5% uranyl acetate in veronal-acetate buffer (pH 6.0), then dehydrated and embedded in Embed-812 resin. Sections were cut on a Leica ultra-microtome with a Diatome diamond knife at a thickness setting of 50 nm, stained with 2% uranyl acetate, and lead citrate. The sections were examined using a Hitachi 7800 and photographed with an AMT ccd camera. To quantify the euchromatin content of nuclei, we used ImageJ software. First, images were converted to 8-bit format, then nuclear area was manually selected, and the threshold was adjusted (using IsoData auto setting).

### Immunofluorescent microscopy
Erythrocytes were isolated from 3-dpf zebrafish embryos (wild-type and miR-144$^{\Delta/\Delta}$) by dissecting their tail using sapphire blade (WPI) and collecting cells in the bleeding buffer (1× PBS containing 2% PBS and 5 mM EDTA). Erythrocytes were washed twice with the bleeding buffer and spread on Superfrost Plus microscope slides using Shandon Cytospin2 Centrifuge at 35× *g* for 5 min and fixed for 20 min in ice-cold methanol. Erythrocytes were incubated for 30 min at room temperature in blocking buffer and stained directly on slides with anti-RNAP II Phosho-Ser2 (Abcam #5095) in the blocking buffer (1X PBS, bovine serum albumin (BSA) 2%, Tween 0.2%) overnight at 4 °C. The slides were then washed three times with 1× PBS, after what anti-rabbit Alexa Fluor 488 secondary antibodies (Jackson Immuno Research Laboratories, #711-545-152) in blocking buffer were added and cells were

incubated for 1 h at room temperature. After that, DAPI was added to cells and incubated for 10 min at room temperature. After final washes, 1× PBS was mounted using Antifade Mounting Media (Vestashield) for fluorescence microscopy imaging. Images were captured using Zeiss Axio Observer Z1 microscope equipped with a digital camera (C10600/ ORCA-R2 Hamamatsu Photonics).

## May–Grünwald–Giemsa staining and N:C ratio

Peripheral blood from different developmental time points was isolated from wild-type, miR-144$^{\Delta/\Delta}$, hmgn2$^{\Delta/\Delta}$, miR-144$^{\Delta/\Delta}$/hmgn2$^{\Delta/\Delta}$ and Tg[mR-144/451::hmgn2-T2A-EYFP] zebrafish by dissecting their tail using sapphire blade (WPI) and collecting cells in the bleeding buffer (1× PBS containing 2% PBS and 5 mM EDTA). Biological diversity is ensured by the fact that each N:C ratio assay analyzes dozens of cells, that come from the pooled blood of ~100 embryos, which in turn are the mixed offspring of multiple breeding pairs. Erythrocytes were spread on Superfrost Plus microscope slides using Shandon Cytospin2 at 35×g for 5 min. Then cells were fixed in iced cold methanol for 5 min and air-dried. Cells were incubated for 5 min in May–Grünwald Stain (Sigma). After that slides were rinsed twice in 1X PBS in transferred into dilute Giemsa solution (1:20) (Sigma) for 20 min. Finally, briefly rinsed in deionized water and air-dried (Sigma). Images were captured using a Zeiss Axio Observer.Z1 microscope with Zeiss ZEN 3.3 Blue software. The quantification of the nucleus-to-cytoplasm (N:C) ratio of erythrocytes was performed in ImageJ. First, images were converted to an 8-bit format, and the threshold was adjusted to select either the area occupied by the nucleus or by the entire cell corresponding to the cytoplasm. Then, images were converted into a binary format, and nuclear and cytoplasmic areas were quantified.

## O-dianisidine staining

To apply mild oxidative stress, live embryos were transferred to water containing 0.003% phenylthiourea (PTU) from 8-hpf until collection time at 48 hpf. Hemoglobin was detected in 2-dpf embryos by incubation for 15 min in the dark in O-dianisidine staining solution (2 mL of water, 2 mL of 0.7 mg/mL O-dianisidine, dissolved in 96% ethanol and protected from light, 0.5 mL of 100 mM sodium acetate, 100 mL of 30% hydrogen peroxide) and then transferred into 1× PBS. After staining, the embryos where imaged in a Discovery V12 Fluorescence Stereomicroscope (Zeiss). The quantification of the area occupied by cells with hemoglobin (O-dianisidine positive) microphotographs of PTU-treated embryos was performed in ImageJ. First, images were converted into 8-bit format. Then, the threshold was adjusted to only select cells with hemoglobin. Next, the image was converted into a binary format and then we quantified the selected area of the selected cells.

## Whole-mount in situ hybridization

Expression of endogenous Hmgn2 was detected by whole-mount in situ hybridization, following the protocols described in ref. 68. Briefly, a 792-bp segment of hmgn2 cDNA was amplified by PCR with oligos that also added a T7 promoter to the amplicon. After agarose gel purification of the PCR amplicon, it was in vitro transcribed with the T7 transcription kit (Invitrogen #AM1333) and the DIG labeling RNA mix/ kit (Roche #39354521) to generate a DIG-labeled antisense probe, complementary to hmgn2 mRNA. One-day-old embryos where fixed with 4% paraformaldehyde (PFA) solution in PBS at pH 7 overnight. After fixation, the embryos were gradually dehydrated in 100% methanol and stored at −20 °C until being used. To initiate the in situ hybridization, embryos are gradually rehydrated with PBS pH 7, and permeabilized by proteinase K digestion (10 μg/ml) at room temperature for 10 min. Proteinase K digestion was stopped by fixation with 4% PFA. Pre-hybridization and overnight hybridization with the DIG-labeled RNA probe was performed as described previously[68]. After washing and blocking, embryos were incubated with an anti-DIG

antibody conjugated to alkaline phosphatase in blocking buffer overnight at 4 °C. Finally, embryos are developed with the staining solution containing NBT and BCIP. Embryos were transferred to a stop solution once the appropriate color intensity was detected. Embryos were imaged in a stereomicroscope Stemi 508, using the Zen suite (Zeiss). Full details of washing steps and buffer compositions can be found in ref. 68.

## ATAC-sequencing

We followed the protocol from ref. 69 with some modifications. Erythrocytes were isolated from WT and miR-144$^{\Delta/\Delta}$ adult zebrafish. Peripheral blood was collected from cut fin of adult fish, or from 2-dpf zebrafish embryos by dissection of caudal vein using sapphire blade, in the collection buffer (1× PBS containing 2% PBS and 5 mM EDTA). Number of blood cells was quantified with Countess Cell Counting Chamber Slides (ThermoFisher) and 50,000 cells were used for every reaction. Collected cells were spun down in the cell collection medium at 500 × g for 5 min at 4 °C. Cell pellets were gently washed with 50 μL of cold 1 × PBS and spun down at 500 × g for 5 min at 4 °C. After removal of PBS, 50 μL of cold cell lysis buffer (10 mM Tris-HCl pH 7.4, 10 mM NaCl, 3 mM MgCl$_2$, 0.1% IGEPAL CA-360) was added, and cells were resuspended by gentle pipetting followed by an immediate spin at 500× g for 5 min at 4 °C. The supernatant was removed, and the purified nuclei were resuspended in the transposition reaction mixture (25 μL 2× TD Buffer, 2.5 μL Tn5 transposase, 22.5 μL Nuclease Free water) and incubated for 90 min at 37 °C. DNA was then purified with Monarch PCR & DNA Cleanup Kit (New England BioLabs). Libraries were prepared using Q5 High-Fidelity 2× Master Mix NEB, M0492) with the following conditions: 72 °C, 5 min; 98 °C, 30 seconds; 15 cycles of 98 °C, 10 s; 63 °C, 30 s; and 72 °C, 1 min. Amplified libraries were purified with Monarch PCR & DNA Cleanup Kit (New England BioLabs) and sequences Illumina NextSeq 2000 system at Boston University Microarray and Sequencing core. ATAC-Seq data was analyzed as described in ref. 69.

## Analysis of candidate genes on erythrocyte morphology

To analyze the effect of overexpression candidate genes (hmgn2, gtf2a1, cbx8a, nap1l4b and dicer1) their mRNAs were injected into one-cell stage wild-type zebrafish embryos (100 pg per embryo). 2-dpf peripheral blood was isolated as described above and analyzed by May–Grünwald–Giemsa staining.

## MicroRNA reporter assay

Full-length hmgn2 3' UTR from zebrafish and its variant lacking two miR-144 targets sites were cloned into pCS2+ after the Nanoluciferase coding sequence. Reporter constructs were linearized with NotI restriction enzyme and in vitro transcribed with mMESSAGE mMA-CHINE SP6 Transcription Kit (Ambion). For the fluorescent miRNA reporter assay, zebrafish embryos were injected with 1 nL of 100 ng/μL of WT 3'UTR hmgn2 or MUT 3'UTR hmgn2 reporter together with firefly luciferase[70] as a control reporter. Synthetic RNA oligonucleotides (IDT) representing miR-144 duplex were annealed by incubation in 1X TE buffer at 90 °C for 5 min and then slow cooled to room temperature. In all, 1 nL of 10 μM miR-144 duplex was injected together with reporters. To quantify the activity of reports groups of five embryos were collected at 8 h post-injection in triplicates and lysed in 100 mL of lysis buffer (Promega). Reporter expression was quantified with the Nano-Glo Dual-Luciferase Reporter Assay System (Promega) in a Synergy H1 Hybrid Multi-Mode Microplate Reader (Thermo).

## Gene expression analysis using RNA-Seq

Peripheral blood cells were isolated from WT and miR-144$^{\Delta/\Delta}$ zebrafish at 3-dpf in biological duplicates. Total RNA was extracted with TRIzol and quantified. Poly(A) mRNA selection and library preparation were performed using NEBNext Ultra II RNA Kit (NEB). Sequencing was

performed at Boston University Microarray and Sequencing core. Sequencing data was analyzed as follows: Transcript-based counts were obtained from Kallisto (v0.46.1) using *Danio rerio* GRCz10 cDNA annotation downloaded from Ensembl. Sleuth (v0.30.0) was used for between sample normalization and to calculate aggregate gene counts (gene tags per million).

## Gene expression analysis using QuantSeq

Peripheral blood cells were isolated from WT and miR-144$^{\Delta/\Delta}$ zebrafish 2-dpf and adult zebrafish in biological triplicates. Total RNA was extracted with TRizol and quantified. Libraries for sequencing were prepared using QuantSeq 3′ mRNA-Seq Library Prep Kit FWD. Quant-Seq libraries were analyzed as described above.

## Single-cell RNA sequencing of the hematopoietic niche of *Danio rerio* whole-kidney marrow and dorsal aorta

Pronephros of adult zebrafish was isolated according to previously described protocol[71]. Fish to be dissected were approximately age-matched, and one male was dissected per genotype. At the time of dissection, fish was euthanized in ice-cold water for 2 min until all breath ceased and its head was removed by making a cut immediately behind the gill operculum using a surgical blade. It was placed on a slightly wet sponge with a notch to maintain the fish on its back. Using a surgical blade, a ventral midline incision was made from anterior to posterior to expose the body cavity, which was held open with forceps. With a second pair of forceps, the body cavity was removed of any sperm, digestive organs, or other tissue. Upon successful cleaning of the body cavity, only the wholekidney marrow (WKM) and dorsal aorta were visible attached to the dorsal body wall of the opened fish. The WKM and attached dorsal aorta were carefully removed using forceps and added to a microcentrifuge tube with 1 mL ice-cold 0.9× PBS containing 2% FBS and 5 mM EDTA, which was then maintained on ice. The sample was resuspended by vortex as pipetting up and down. The entire sample was pipetted onto a 40-μm mesh (Falcon) over a 50 mL Falcon tube and crushed with the plunger of a 1-mL syringe. Cells were pelleted by centrifugation at 200×g for 5 min at room temperature. Cells were resuspended in 1 mL of ice-cold 0.9× PBS containing 2% FBS and 5 mM EDTA and transferred to a 1.5-mL microcentrifuge tube. The cells were pelleted once again by centrifugation at 200 × g for 5 min. Upon aspiration of the supernatant, the pellet was resuspended in 500 μL of ice-cold 0.9× PBS containing 2% FBS and 5 mM EDTA and passed through a 35-μm mesh cell strainer cap into a 5 mL polystyrene round-bottom tube (Falcon). To the resuspension, 1 μL of 1 mM TO-PRO-3 Iodide in DMSO (Thermo) was added to stain dead cells, and 1 μL of DRAQ5 (Invitrogen) was added to stain viable cells. A Beckman Coulter MoFlo Astrios cell sorter was used to gate-select for viable cells of the hematopoietic niche, while excluding non-hematopoietic kidney cells. These cells were sorted out into 1.5-mL microcentrifuge tube containing 50 μL of ice-cold 0.9× PBS containing 2% FBS and 5 mM EDTA. The sorted samples were kept on ice and a fraction was used to count the cells in a hemocytometer. The sample was then processed for single-cell RNA sequencing. Single-cell sequencing was performed using 10x Genomics platform at Boston University Microarray and Sequencing core. scRNA-Seq data was analyzed as follows: Reads were aligned to the zebrafish genome (GRCz11) with STAR (v2.7.9). Ensembl (v104) gene annotations were downloaded as a GTF file and used for the analysis. Results files in h5ad format were read into Python and labeled as WT or miR-144 and then combined into a single Anndata object. Scanpy was used for normalization and filtering. After creating the UMAP and clustering it was observed that one outlier cluster was enriched for genes expressed in sperm and was removed as it was likely contamination on sample collection. Clusters were annotated with the set of marker genes used in ref. 72 and progenitor cell and erythrocyte clusters were further analyzed to compare the expression of *hmgn2* in the wild-type and miR-144 mutant genotypes. Expression of marker

genes and *hmgn2* is shown as a heat map over the UMAP plot, where the color scale represents the natural log of normalized read counts +1, or ln(cpm+1). Developmental trajectories were established using scVelo[37] and plotted by applying partition-based graph abstraction (PAGA)[38].

## Cloning, lentivirus production, and infection

We designed an Hmgn2–3′UTR reporter based on the pHAGE lentiviral vector that has been previously described[73,74] DNA sequences encoding human Hmgn2 coding region, EYFP and either wild-type (*hs*Hmgn2-EYFP-3′UTR-WT) or mutant lacking miR-144 target sites (*hs*Hmgn2-EYFP-3′UTR-MUT) 3′UTRs were ordered directly as gBlocks (IDT) and inserted into pHAGE vector at BamH1 and Not1 sites. Lentiviruses were produced using a five-plasmid transfection system in 293T packaging cells as previously described[73]. Supernatants were collected every 12 h on 2 consecutive days starting 48 h after transfection, and viral particles were concentrated by centrifugation in SW 32 Ti rotor (Beckman Coulter) at 46,417 × g for 1.5 h at 4 °C. Immortalized iPSC-derived erythroid cells were infected with 15 μl of concentrated virus in the presence of polybrene (5 μg/ml). The medium was replaced after 16 h with base Serum-Free Expansion Medium I (SSI) supplemented with 2 mM of L-glutamine, and 100 μg/mL of primocin 40 ng/mL of IGF1, $5 \times 10^{-7}$ M of dexamethasone, and 0.5 U/mL of hEPO.

## Erythroid differentiation of human iPSCs

SS-14 cells of female origin were obtained from a library of human sickle cell disease induced pluripotent stem cells (iPSC) thoroughly described in[75]. Differentiation of these iPSCs faithfully recapitulate human erythropoiesis in vitro[76]. Hematopoietic differentiation from iPSCs to hematopoietic stem and progenitor cells (HSPCs) was induced according to ref. 41. HSPCs were then transferred into SSI medium (see below) to induce erythroid specification. After 5 days into SSI medium, when the cells reached a stable proerythroblast stage, a lentiviral vector carrying large T antigen was introduced to immortalize cells. Immortalized cells were subsequently infected with lentivirus encoding *hs*Hmgn2-EYFP-3′UTR-WT, *hs*Hmgn2-EYFP-3′UTR-MUT or EYFP alone. To induce further erythroid differentiation from what we consider baseline ("day 0"), cells were cultured for 2 more days in SSI media ("day 2"), followed by 5 days into SSII media under hypoxic conditions (see below, "day 7"). SSI and SSII media are parts of a 2-step suspension culture system with a base medium consisting of StemSpan Serum-Free Expansion Medium II (StemCell technologies), 2 mM of L-glutamine, and 100 μg/mL of primocin. To create SSI medium, this base was supplemented with 100 ng/mL of human stem cell factor, 40 ng/mL of IGF1, $5 \times 10^{-7}$ M of dexamethasone, and 0.5 U/mL of hEPO and cells were cultured at 37 °C in normoxic, 5% carbon dioxide conditions. SSII medium is created by adding 4 U/mL of hEPO to the base medium. Cells were cultured in hypoxic conditions during the maturation phase in SSII.

## cDNA preparation and qPCR

RNA from fish embryos was extracted using the Trizol (Qiagen) and DNase treated using a DNA-free kit (Ambion). Complementary DNA (cDNA) was prepared from total RNA (5 μg) by reverse transcription using LunaScript® RT SuperMix Kit (NEB). qPCR reactions contained a pair of oligonucleotides priming EYFP sequence (Supplementary Data 1) and Power SYBR Green Master Mix (Thermo) and were performed in a ViiA7 Real-Time PCR System (Applied Biosystems). Data were normalized to *GAPDH* mRNA amplification. For the qPCR expression analysis of globins in human cells, predesigned TaqMan primers (Applied Biosystems) were used in conjunction with Taqman Universal Master Mix II (Applied Biosystems; #4440038) for quantitative reverse transcription PCR analysis on the StepOne/QuantStudio 6 Flex Real-Time PCR Systems (Applied Biosystems). A combination of the housekeeping genes *GAPDH* and *ACTIN* was used as an endogenous

control. Data were analyzed using the ddCT method. These normalized fold expression values for all time points were then divided by the day-0 values to obtain relative fold expression compared with day 0 of erythroid differentiation.

## Flow cytometry analysis

Cells were stained on ice for 25 min using the following antibodies: BD Horizon™ BV421 Mouse anti-Human CD235a antibody (#562938) or PE Mouse anti-Human CD235a antibody (#555570), Flow cytometry was conducted on a Stratedigm S1000EXI. FlowJo v8.7 (FlowJo, LLC) software was used for analysis, and FACS plots shown represent live erythroid cells based on side-scatter/forward-scatter gating.

## Small RNA northern blotting

Total RNA was extracted using Trizol (Invitrogen), quantified (we used 10 µg of total RNA per lane) and resuspended in formamide. Loading buffer 2× (8 M urea, 50 mM EDTA, 0.2 mg/ml bromophenol blue, 0.2 mg/ml xylene cyanol) was added, and the samples were boiled for 5 min at 95 °C. miRNAs were separated in 15% denaturing urea polyacrylamide gel in 1× TBE and then were transferred to a Zeta-Probe blotting membrane (Bio-Rad) using a semi-dry Trans-Blot SD (Bio-Rad) at 20 V (0.68 A) for 35 min. Membranes were UV cross-linked and pre-hybridized with ExpressHyb Hybridization Solution (Clontech) for 1 h at 50 °C. Membranes were blotted with 5′ $^{32}$P-radiolabelled DNA oligonucleotide probes at 30 °C overnight. Membranes hybridized with oligonucleotide DNA probes were washed at room temperature with 2× SSC/0.1% SDS followed by 1× SSC/0.1% SDS for 15 min. The blots were exposed to a phosphorimaging screen for 1–3 days. The signal was detected using the Typhoon IP phosphorimager (GE Healthcare Life Technologies) and analyzed using the ImageQuant TL software (GE Healthcare).

## Preparation of radiolabeled probes

Radiolabeled DNA probes were prepared according to the StarFire method. Oligos carrying specific DNA sequence complementary to the miRNA of interest were annealed to the universal oligo (5′-TTTTTTTTTTT666G6(ddC)-3′, where "6" corresponds to a propyne dC modification) via complementary hexamer sequence. Annealed duplexes are then labeled with α-$^{32}$P-dATP (6 µL of 10 mCi/mL stock) using the Klenow fragment of DNA polymerase. Reaction was stopped by adding 40 µL of 10 mM EDTA solution to 10 µL of reaction. Then labeled oligos were purified using Micro-Spin G25 columns (GE HealthCare). 3,000,000 cpm of the P$^{32}$ labeled StarFire probes were used to probe the membrane.

## Immunoprecipitation

FLAG-Dicer1 overexpressed in embryos was pull-down using FLAG M2 magnetic beads (Millipore Sigma, #M8823). Briefly, 150 embryos collected at 6-hpf were lysed with 400 µL NET-2 buffer (100 mM Tris-HCl pH 7.5, 150 mM NaCl, 0.05% NP-40) and protease/phosphatase inhibitor cocktail (ThermoFisher) by pipetting up and down for several times. The supernatant was collected by centrifugation at 16,000 × $g$ for 10 min at 4 °C and then incubated with 100 µL FLAG M2 beads (Sigma-Millipore) for 1 h at 4 °C. The beads were washed three times with 400 µL NET-2 buffer. In total, 40 µL of 1.5x Laemmli buffer was added to the beads and boiled at 95 °C for 10 min to extract the protein.

## Western blotting

Proteins were resolved using precast NuPAGE Novex 8% Bis-Tris gel plus (Invitrogen) at room temperature and constant voltage (100 V) in 1× MOPS Running buffer and transferred to a nitrocellulose membrane (0.45 µm, Bio-Rad) using iBlot2 system (Invitrogen). After transfer, membranes were blocked in a blocking buffer (5% non-fat dry milk in TBS-T) for 1 h at room temperature. Membranes were incubated with

primary antibody overnight at 4 °C. After that, membranes were washed three times with TBS-T and then incubated with secondary antibody for 2 h at room temperature. Primary and secondary antibodies were diluted in TBS-T as follow: anti-HA (Millipore Sigma, #H6908, 1:1000), anti-Actin (Millipore, #MAB1501R, 1:1000), anti-FLAG M2 (Millipore Sigma #F7425, 1:1000), IRDye 800CW Goat anti-mouse (Li-COR, #925-32210, 1:10,000), IRDye 680RD Goat anti-rabbit (Li-COR, #926-68071, 1:10,000), IRDye 800CW Goat anti-rabbit (Li-COR, #926-32211, 1:10,000), All membranes were scanned using Odyssey Scanner (Li-COR) and band intensity was quantified with Odyssey software.

## Statistical analysis

Bar plots indicate the mean, and the associated error bars represent the standard error of the mean. Box plots enclose the 25th to 75th percentile, and the whiskers delimit 5th to 95th percentiles, with the median indicated as a black line across the box. To determine the statistical significance of differences between two groups, we conducted two-tailed unpaired $t$ test. For multiple groups, we conducted a one-way ANOVA test followed with a Dunnett's multiple comparisons test. For all tests, the significance threshold was set for $P = 0.05$. qPCR data for globin genes expressed in differentiating iPSC were analyzed fitting a mixed model with a Geisser–Greenhouse correction as implemented in GraphPad Prism 10.0. This mixed model uses a compound symmetry covariance matrix and is fit using Restricted Maximum Likelihood (REML). This analysis tests the null hypothesis that all genotypes have the same population means. All statistical analyses were conducted with GraphPad Prism, version 10.0.3. Figure legends describe the sample size and statistical analysis done in each corresponding plot.

## Reporting summary

Further information on research design is available in the Nature Portfolio Reporting Summary linked to this article.

## Data availability

All sequencing data were deposited to NCBI as SRA under the following Bioproject identifiers: (1) ATAC-seq data: PRJNA1007738, (2) RNA-Seq, scRNA-Seq and QuantSeq data: PRJNA1010662. Raw images are deposited in Figshare under https://doi.org/10.6084/m9.figshare.25552233 [https://figshare.com/articles/figure/NCOMMS-23-34907A/25552233]. Reasonable requests for resources and reagents should be directed to and will be fulfilled by the Lead Contact, Daniel Cifuentes (dcb@bu.edu). Source data are provided with this paper.

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

## Acknowledgements

The authors thank A. Grishok, X. Varelas, M. Blower, M. Garcia-Marcos and L. Zon for fruitful discussions and access to instruments and microscopes. The authors thank Yuriy Alekseyev, Joshua Campbell, and the Boston University Microarray and Sequencing Resource for conducting sRNA-Seq and scRNA-Seq sequencing and analysis. We thank Nicki Watson (Harvard) for conducting Transmission Electron Microscopy. The authors thank Boston University Flow Cytometry Core for cell sorting. We thank Charith Wijeyesekera for the help with iPSCs experiments. We thank Andrew Tilston-Lunel for taking photos of *miR-144/451::EYFP/gata1::dsRed* transgenic. Cartoons in Figs. 2B, 2F, 4C, and 5A are reprinted from ref. 17, with permission from Elsevier. This work was supported by the National Institute of Health (US) grant R01GM130935-03 (awarded to D.C.), a seed grant for scRNA-Seq awarded by Boston University Genome Science Institute to D.C. and an RNA-sequencing award awarded by Boston University Genome Science Institute to D.A.K.

## Author contributions

D.A.K. designed, performed, and analyzed all the experiments and wrote the draft of the manuscript. L.F. analyzed ATAC-sequencing data. A.M.M. performed whole-mount in situ hybridization. I.A.W. performed sample preparation for single-cell analysis of zebrafish pronephros. I. tested Dicer expression and activity. N.S. cloned candidate genes and helped with the generation and maintenance of zebrafish lines. S.M. analyzed Quant-sequencing, RNA-sequencing data and single-cell sequencing data. K.V. and G.M. designed and helped with human iPSCs experiments and erythroid differentiation. D.C. conceived and supervised the project, performed experiments, analyzed the data, acquired funding, and wrote the manuscript.

## Competing interests

The authors declare no competing interests.
