## [Peer Review File · Nature Communications]

The miR-144/Hmgn2 regulatory axis orchestrates chromatin organization during erythropoiesisREVIEWER COMMENTS

Reviewer #1 (Remarks to the Author):

In this manuscript, Kretov and colleagues uncover a new relationship between miR-144 and one of its targets, Hmgn2, in the process of chromatin remodelling during haematopoiesis. By using several well thought out and elegant experiments, they demonstrate that miR144 activity is required to aid in chromatin compaction during haematopoiesis and show conservation of this mechanism in human using an iPSC differentiation system in vitro. Overall, this is a well-executed study that makes good use of various state-of-the-art technologies to demonstrate the existence of a miR144/Hmgn2 axis in regulating chromatin accessibility. I believe that addressing a few points as set out below would help improve this manuscript for publication in Nature communications.

Major comments

1. Some data would benefit from a clearer presentation in the text and description in the figures/figure legends (e.g. current fig 2A, fig 2G, figure 4D, fig S4B).
2. In the results section, there are several instances of figure panels being referred to out of order (e.g. 5C first, then 5A and B); I'd suggest to either modify the text or change the figures to ensure they flow better with the text. For example Fig 5C could be supplementary, or moved to 5A instead.
3. The interpretation of the scRNAseq expression data for hmg2 from the adult marrow is overstated; it's not clear from the UMAP presented that they can extrapolate or infer a trajectory of differentiation, and they only show either lineage-specific gene expression (supplementary) or hmg2 expression (figure 2G). It would be best to use an algorithm for trajectory inference to demonstrate the correlation between hmg2 expression and erythroid differentiation and support their claim of inverse correlation with differentiation.
4. You need to quantitate the o-dianisidine staining (figure 4E) to back up the mild anemia claim and provide some evidence that PTU causes that much oxidative stress. Many people use it for bleaching pigment from 24hpd onwards at the stated concentration, so this would be important to clarify for others in the field.

Minor comments

5. Pp5, lines 7-9 – it's not immediately obvious which experiment was performed to look at phosphorylated PolII
6. Pp5, lines 23, 24 – the use of the term 'stabilized' and 'destabilized RNAs is a bit confusing nomenclature – do you mean enriched and depleted?
7. Pp7, line 15-16 – where it reads 'in for miR-144-v1' should it be 'for miR-144-v1'?
8. Pp8, lines 4-6 – the claims of differentiation stages for erythroid cells/progenitors need to be better supported with methods of trajectory inference for single cell data.
9. On that note, there's no mention in either the methods or results of the number of kidney marrow cells that were used for the scRNAseq, or the number of samples of each genotype, etc.
10. The cross between the mir144/451::EYFP line with the gata1:Dsred line is a great experiment – did you only do this at day 1? It would be interesting to see how it progresses in later stages (say up to day

5).

11. Pp9, line 17 – transgenic embryos at 1dpf is rather vague as many things happen between 24hpd and 48hpf – what stage were the embryos at when they were imaged?

12. Pp9, line 22 – it's unclear what the qPCR analysis was. Was it on gfp?

13. Pp10 – can't see a reference to panel 4B

14. The log₂ peak coverage correlation as a representation of changes in ATAC peaks (fig 4F, S4B) is not particularly good (in my view) to represent gains/losses of peak accessibility – figure 1H is much clearer in that respect. The description/labelling of those panels is quite thin as well, which does not help.

15. Pp11, lines 5-7 – reducing levels of hemgn2 rescue the N:C ratio but not to the extent that they rescue chromatin accessibility? How can you explain this discrepancy?

16. Did you try to mimic the hemgn2 KO by injecting miR-144 mimics?

17. Pp15, lines 29, 30 – two references missing

Reviewer #2 (Remarks to the Author):

Kretov et al use a miR-144 zebrafish mutant to link their defect in nuclear chromatin compaction with loss of miR-144-dependent negative regulation of Hmgn2.

Major comments

1. The abstract claims to have shown a functional pathway in both zebrafish and human systems. While the studies in zebrafish are genetically comprehensive and elegant and have generated functional results supported by analytical statistics, the studies in human iPCSs are much weaker. The evidence that the pathway that has been dissected very comprehensively in zebrafish systems is conserved in mammals is confined to: (1) the Hmgn2 3'UTRs of many vertebrates carry miR-144 site(s); (2) the final results section based on iPCS erythroid differentiation (p11 and Fig 5). Regarding this iPCS experiment, the essentially qualitative findings of the p11 text reflect the fact that the functional results of Fig 5G and H are not robustly supported by analytical statistics. The histogram of Fig 5G is from one sample in one experiment; there is no evidence the experiment was replicated. The description of the difference in Hmgn2 3'UTR mutant sample ("delayed and reduced") is not supported by any analytical statistical proof of delay or reduction. Again, these data appear to all come from one experiment with only technical replicates.

1a. In the narrative sequence of the paper, the "conserved across vertebrates" claim on p 8, line 12 is premature. At this point in the narrative, only the conservation of miR-144 binding sites is demonstrated in fish and mammals; up to that point, all functional data derive from zebrafish experimentation.

2. Descriptive and analytical statistics and reproducibility.

There is no statistical section in the methods section, so information is provided only in figure legends.

2a. Were the unpaired t-tests one- or two-tailed?

2b. What is meant by "ordinary one-way ANOVA"?

In Fig 1B, a simple one-way ANOVA at each of three time point can be considered appropriate (although a 2-way ANOVA could have been considered as there are two independent variables (genotype and timepoint) influencing the dependent variable (N:C ratio).

However, in Fig 2B, did the “ordinary one-way ANOVA” include Dunnett’s post hoc test (as the p-value bracketing would suggest).

2c. Why show mean \pm SD for Fig 2F when other results are mean \pm SEM?

2d. Why are box plots sometimes delineating the 5th and 95th percentiles, and other times the 10th and 90th percentiles?

2e. It is general unclear how replicated key results are. For example, in the N:C ratios (Fig 1B,2B, 3G, 4F, were all the cells from one single preparation, which would mean that though there are multiple cells, it is only one biological experiment? Were these preparations prepared several times?

2f. How many embryos in each group (x with phenotype /y total) demonstrate the WISH phenotypes shown by 2 examples/group in Fig 2D?

2g. Again, for Fig 4F, shouldn’t this be a one-way ANOVA with Dunnett’s post hoc test, rather than an “ordinary one-way ANOVA”?

2h. Fig S1B. The error bars mentioned in the legend are missing from the figure.

3. In Figure 3H, I believe that the first and third cell of the five cells arrayed in the TgHmgn2 group is the same cell (and possibly the same image), based on the characteristic indentations in the external cytoplasm, the pattern of punctate lucencies in the nucleus, and a lighter staining cytoplasmic area adjacent to the nucleus from 12 to 3 o’clock.

This observation made me look carefully at the other erythrocyte images, and I also believe that in Fig 2C, for the Gtf2a1 group, the top cell and the second bottom cell are also the same cell (and possibly the same image), based on the eccentric nuclear shape with its broad protrusion at 9 o’clock, a linear lucency in the cytoplasm to the right of the nucleus, and the distribution of thicker darker areas in the cytoplasmic membrane.

While these might individually appear to be minor errors (only 4 examples rather than 5 are provided), the fact that it has occurred twice undermines confidence in all the quantitative N:C ratio data. Within these datasets, some datapoints of equal value are plotted among the individual values shown in the >90 (or >95) and <10 (or <5) percentile ranges, accepting that the visual resolution in the plot is such that data points might only appear to be equal. The duplicate cells/images within the representative examples suggests that all the N:C image datasets that have been quantified need to be audited for cells photographed twice or for duplicate images.

Also relevant to this point is the question about replication of these experiments in point 2e.

4. I did not find the schematics helpful, given the compounding negatives. I recognise this might be an individual thing. For example, in Fig3I, in the WT scenario, the negative regulation by miR-144 of Hmgn2 leads to nuclear condensation. However, when miR-144 is removed and Hmgn2 action is increased, the result is LESS Nuclear condensation (although the blue arrow is bigger and it is point to the words “Nuclear condensation”). The schematics are more diagrams of the experimental intervention, rather than schematics indicating the outcome of the experimental intervention – consider, whether the term

should be “Decondensation”, or adding an up or down arrow, to help.

5. Fig 2B, Dicer1 result. This is a negative result from Dicer1 mRNA injection – it needs to be more firmly technically documented, given the significance made of the finding (p6 lines 17-23). How was it proven that intact Dicer1 mRNA was synthesised and delivered? Were the mRNAs verified as intact at delivery by gel electrophoresis? Was a dose of 100 pg sufficient for this mRNA preparation? One standard way to verify intact RNA delivery is to co-inject all test samples with a fluorophore-encoding mRNA and looking for its expression, and also have a control group of this tracking mRNA only, to control for non-specific RNA effects.

Minor points

1. p15 lines 29-30. Missing references
2. p16 line 7. Spelling of “glutaraldehyde”
3. p17 line 30. Spelling of “appropriate”
4. I expect the journal will require a scale bar for the erythrocyte pictures, at least initially in Fig 1A and C.
5. Fig 5 legend mentions HBA and HBE, but panels are labelled HBA, HBB and HBG, presumably meaning alpha, beta and gamma.
6. The “cDNA preparation and qPCR” section (p21) does not define the primers used for the zebrafish genes.

REVIEWER COMMENTS

We want to take this opportunity to collectively thank the reviewers for their comments, which have guided us to further improve the rigor and clarity of the manuscript.

The original comments of the reviewers are listed in **black**; our responses are written in **blue**.

Reviewer #1 (Remarks to the Author):

In this manuscript, Kretov and colleagues uncover a new relationship between miR-144 and one of its targets, Hmgn2, in the process of chromatin remodeling during haematopoiesis. By using several well thought out and elegant experiments, they demonstrate that miR144 activity is required to aid in chromatin compaction during haematopoiesis and show conservation of this mechanism in human using an iPSC differentiation system in vitro. Overall, this is a well-executed study that makes good use of various state-of-the-art technologies to demonstrate the existence of a miR144/Hmgn2 axis in regulating chromatin accessibility. I believe that addressing a few points as set out below would help improve this manuscript for publication in Nature communications.

We thank the reviewer for recognizing our efforts in conducting “*well thought out and elegant experiments*” and considering our work a “*well-executed study*”. We are glad to read that the reviewer agrees with our main conclusion based on the experimental data provided— i.e., the demonstration of the existence of a miR-144/Hmgn2 regulatory axis regulating chromatin accessibility during hematopoiesis. Nevertheless, we have taken very seriously the few points raised by this reviewer, which we have addresses to the best of our abilities as described below.

Major comments

1. Some data would benefit from a clearer presentation in the text and description in the figures/figure legends (e.g. current fig 2A, fig 2G, figure 4D, fig S4B).

This point is well taken. We have now added better descriptions of the experiments and data described for each one of the figures mentioned:

- In **Figure 2A**, both in the main text (**p6, lines 19-20**) and the corresponding figure legend.
- For **Figure 2G**, we have reinforced our analysis of scRNA-seq data, including a new developmental trajectory analysis between clusters using scVelo (PMID: 32747759) and added clarifications in the main text (**p8, lines 19-20**) and the figure legend.
- Regarding **Figure 4D** and **Figure S4B**, in the initial version of the manuscript they showed our ATAC-Seq data as a scatter plot. We agree with the reviewer that this may not be the standard way in the field to represent ATAC-Seq data. So now in the new version of the manuscript. **Figure 4D** (now **new Figure 4E**) and **Figure S4B** represent the chromatin accessibility data with the more familiar heat map plots.

2. In the results section, there are several instances of figure panels being referred to out of order (e.g. 5C first, then 5A and B); I'd suggest to either modify the text or change the figures to ensure they flow better with the text. For example Fig 5C could be supplementary, or moved to 5A instead.

We agree with the reviewer that the flow of the manuscript is important, and this also applies to the order of the figures. We have reordered the text and the figure panels that the reviewer suggested. Now all figure panels are referenced in the text and in order of appearance.

3. The interpretation of the scRNAseq expression data for hmgn2 from the adult marrow is overstated; it's not clear from the UMAP presented that they can extrapolate or infer a trajectory of differentiation, and they only show either lineage-specific gene expression (supplementary) or hmgn2 expression (figure 2G). It would be best to use an algorithm for trajectory inference to demonstrate the correlation between hmgn2 expression and erythroid differentiation and support their claim of inverse correlation with differentiation.

As requested by the reviewer, we have now determined the best fitted developmental trajectories using the algorithm scVelo (PMID: 32747759) and used partition-based graph abstraction (PAGA) (PMID: 30890159) to plot the results as directional black arrows that indicate the inferred developmental path between clusters in **Figure 2G** and **Figure S3B**. This new analysis clearly indicates the developmental paths taken by hematopoietic progenitors to differentiate into mature erythrocytes. Accordingly, we have expanded the main text and the methods section to include details about our trajectory inference analysis (please see **p8, lines 6-9**). Overall, this new analysis supports our conclusion that miR-144 mutant erythrocytes do not mature as efficiently as the wild-type cells and that they express higher levels of Hmgn2.

4. You need to quantitate the o-dianisidine staining (figure 4E) to back up the mild anemia claim and provide some evidence that PTU causes that much oxidative stress. Many people use it for bleaching pigment from 24hpd onwards at the stated concentration, so this would be important to clarify for others in the field.

In response to the reviewer prompts, now we have quantified the O-dianisidine staining of **Figure 4E** (now **new Figure 4H**). The quantification of the staining is shown in **Figure 4G**. In addition, we have added a paragraph and citations in the main text (**p11, lines 13-17**) that clarifies this use of PTU as oxidative stress agent.

Briefly, we agree with the reviewer that PTU is widely used to inhibit pigmentation of zebrafish embryos with the goal to obtain nice images without being obscured by melanocytes. What researchers may not be aware is that the use of PTU is a well-established paradigm to induce mild oxidative stress during hematopoietic studies. Yu et al, (PMID:20679398) used PTU to exacerbate the effects of miR-451 inhibition with morpholinos. While wild-type embryos are barely affected, miR-451 morphant embryos become anemic. Interestingly, treatment with the antioxidant N-acetyl cysteine rescues the PTU-induced hematopoietic defects. We already used PTU in a previous manuscript in a similar way to study miR-451 knock-out fish (PMID:32191872). PTU also induces oxidative stress in other systems, like neurons and glioblastoma cells (PMID:10987861, 16447258). The mechanism of action is due in part to the PTU-mediated inhibition of dopamine b-hydroxylase and tyrosinase. This blockade results in the increase of cytotoxic intermediate products like quinones, semiquinones and reactive oxygen species (PMID:10987861).

Minor comments

5. Pp5, lines 7-9 – it's not immediately obvious which experiment was performed to look at phosphorylated PolII.

We concur with the reviewer that the main text did not have enough information to convey the relevance or the methodology of evaluating RNAP II phosphorylation. We have now improved the text by explicitly indicating that the fraction of RNAP II phosphorylated at the serine in position 2 is actively engaged in productive elongation and that we detect the phosphorylation by immunofluorescence using a specific antibody.

6. Pp5, lines 23, 24 – the use of the term ‘stabilized’ and ‘destabilized RNAs is a bit confusing nomenclature – do you mean enriched and depleted

We agree with the reviewer that “enriched” and “depleted” are more encompassing terms to describe mRNA levels. We have now amended the main text accordingly (**p5, lines 25-26**) and the legend in the plot of **Figure 2A**.

7. Pp7, line 15-16 – where it reads ‘in for miR-144-v1’ should it be ‘for miR-144-v1’?

Thank you for pointing us to this grammatical error. We have corrected it and now the main text reads: “...has two predicted 7mer-m8 target sites for miR-144-v1 in its 3’UTR”

8. Pp8, lines 4-6 – the claims of differentiation stages for erythroid cells/progenitors need to be better supported with methods of trajectory inference for single cell data.

As discussed in point 3, we now have conducted a trajectory inference analysis and it is reflected in the main text and the methods. Specifically, we determined the best fitted developmental trajectories using scVelo (PMID: 32747759) and used partition-based graph abstraction (PAGA) (PMID: 30890159) to plot the results as directional black arrows that indicate the inferred developmental path between clusters in **Figure 2G** and **Figure S3B**.

9. On that note, there’s no mention in either the methods or results of the number of kidney marrow cells that were used for the scRNAseq, or the number of samples of each genotype, etc.

We thank the reviewer for pointing to this omission. Now, **Figure S3C** contains a table breaking down the total number of cells per genotype and per cluster. In summary, we analyzed 2,034 wild-type cells and 1,701 miR-144 mutant cells, distributed over 6 clusters. In addition, we updated the methods section to indicate that we dissected one fish per genotype.

10. The cross between the mir144/451::EYFP line with the gata1:Dsred line is a great experiment – did you only do this at day 1? It would be interesting to see how it progresses in later stages (say up to day 5).

We did not show the overlap of the fluorescent reporter at later stages because the expression of EYFP driven by miR-144/451 promoter decreases after 48hpf. This result fully recapitulates the miR-144/451 expression dynamics that we determined in a previous paper (see attached figure from PMID: 32191872), where we observe that the miRNA cluster expression peaks at 48hpf.

11. Pp9, line 17 – transgenic embryos at 1dpf is rather vague as many things happen between 24hpd and 48hpf – what stage were the embryos at when they were imaged?

We imaged the embryos in **Figure 3E** at 30-hpf. We concur with the reviewer that indicating the developmental time point in hours post-fertilization rather than days post-fertilization is more accurate. This more accurate time is now reflected in the main text (**p9, line 27**). We also converted dpf to hpf in **Figure 1B** to be more precise.

12. Pp9, line 22 – it's unclear what the qPCR analysis was. Was it on gfp?

We apologize for the lack of detail. Now, we have amended the main text (**p9, line 33**) and the methods section to indicate that we determined the expression levels of the transgenic cassette using a pair of oligonucleotides specific for the EYFP embedded in the Hmgn2-T2A-EYFP transgenic cassette. In addition, we have added a new supplementary table (**Table S1**) that includes all the oligonucleotide sequences used in the manuscript.

13. Pp10 – can't see a reference to panel 4B

Please excuse the omission. Now the text incorporates the reference to **Figure 4B**, that show how loss of Hmgn2 does not impair gross morphological development in zebrafish embryos.

14. The log₂ peak coverage correlation as a representation of changes in ATAC peaks (fig 4F, S4B) is not particularly good (in my view) to represent gains/losses of peak accessibility – figure 1H is much clearer in that respect. The description/labelling of those panels is quite thin as well, which does not help.

We concur with the reviewer on the point that heat maps are better at showing differences in chromatin accessibility and are the standard in the field. Accordingly, we have substituted the scatter plots in **Figure 4F** (now **new Figure 4E**) and **S4B** for heat maps to represent ATAC-Seq data, like we did for **Figure 1H**. With this changes we also standardized how we present ATAC-Seq data across the manuscript.

15. Pp11, lines 5-7 – reducing levels of hemgn2 rescue the N:C ratio but not to the extent that they rescue chromatin accessibility? How can you explain this discrepancy?

In this manuscript we show that there are multiple chromatin factors that are enriched upon the loss of miR-144 (**Figure 2A**) and that their overexpression has an impact in the N:C ratio (**Figure 2B and C**). In the double mutant miR-144/Hmgn2 we are interrogating the effects of Hmgn2, but there are still other chromatin factors that are expressed above normal levels that may contribute to this pervasive open chromatin status detected by ATAC-Seq. We state this hypothesis in the main text (**p9, lines 15-11**).

16. Did you try to mimic the hemgn2 KO by injecting miR-144 mimics?

We did not try to mimic the loss-of-function phenotype induced by Hmgn2 deletion by injection of miR-144 duplex in wild-type embryos because the results would not be conclusive. This is because wild-type embryos already express endogenous miR-144 that represses Hmgn2. At the same time, the exogenous miR-144 will also repress Dicer, which in turn will facilitate miR-451 and enhance erythrocyte differentiation (PMID:32191872). It will be out of the scope of this manuscript to conduct the additional experiments that would tease apart what portion of the miR-144 injection phenotype is due to repression of *hmgn2*, Dicer or the other chromatin factors that we tested (**Figure 2B and C**).

17. Pp15, lines 29, 30 – two references missing

Thank you for detecting this omission. Now the two references are cited correctly.

Reviewer #2 (Remarks to the Author):

Kretov et al use a miR-144 zebrafish mutant to link their defect in nuclear chromatin compaction with loss of miR-144-dependent negative regulation of Hmgn2.

Major comments

1. The abstract claims to have shown a functional pathway in both zebrafish and human systems. While the studies in zebrafish are genetically comprehensive and elegant and have generated functional results supported by analytical statistics, the studies in human iPCSs are much weaker. The evidence that the pathway that has been dissected very comprehensively in zebrafish systems is conserved in mammals is confined to: (1) the Hmgn2 3'UTRs of many vertebrates carry miR-144 site(s); (2) the final results section based on iPCS erythroid differentiation (p11 and Fig 5). Regarding this iPCS experiment, the essentially qualitative findings of the p11 text reflect the fact that the functional results of Fig 5G and H are not robustly supported by analytical statistics. The histogram of Fig 5G is from one sample in one experiment; there is no evidence the experiment was replicated. The description of the difference in Hmgn2 3'UTR mutant sample ("delayed and reduced") is not supported by any analytical statistical proof of delay or reduction. Again, these data appear to all come from one experiment with only technical replicates.

We are grateful for the kind words of the reviewer that defines our zebrafish work as "*genetically comprehensive*" and "*elegant*". At the same time, we hear the reservations of the reviewer regarding the conclusions from the experiments iPCS and we have gone to great lengths to address them. More specifically, we have now performed experiments to include additional biological replicates to support our flow cytometry results (**new Figure S5**), which consistently show that expression of Hmgn2 with the mutant 3'UTR impairs CD235 expression, a proxy for erythrocyte maturation. We have softened the claim about Hmgn2 with the wild-type 3'UTR because the effects are less consistent. This result agrees with the fact that the cells with the wild-type and mutant 3'UTR over-express Hmgn2 2.5 and 16-fold above the endogenous levels, respectively, and hence the effect will be more marked with the mutant 3'UTR.

We also now provide statistical analysis for the differential expression of globin genes (**Fig. 5H**). We could apply a two-way ANOVA due to two random missing values. So instead, we analyzed the data by fitting a mixed model with a Geisser-Greenhouse correction as implemented in GraphPad Prism 10.0. This mixed model uses a compound symmetry covariance matrix and is fit using Restricted Maximum Likelihood (REML). This analysis rejects the null hypothesis that all genotypes have the same population means of globin expression for HBA ($p < 0.0001$), HBG ($p = 0.0417$), and HBE ($p = 0.0015$), but not HBB ($p = 0.3536$), suggesting that the erythrocyte maturation is indeed impaired. Considering these results, we have softened the claim in the main text and changed "delayed and reduced" to "impaired" erythrocyte maturation (**p12, lines 28-33**).

1a. In the narrative sequence of the paper, the "conserved across vertebrates" claim on p 8, line 12 is premature. At this point in the narrative, only the conservation of miR-144 binding sites is demonstrated in fish and mammals; up to that point, all functional data derive from zebrafish experimentation.

We concur with the reviewer that up to this point of the manuscript, the experimental validation of hmg2 as a target of miR-144 is only done in zebrafish, and for the rest of vertebrates we only list the occurrence of miR-144 target sites in their 3'UTR sequence. Accordingly, we have amended the main text (**p8, lines 22-24**) to remove the statement about conservation, which is brought up only later when the relevant data to support the statement is presented (**p12, line 19**).

2. Descriptive and analytical statistics and reproducibility. There is no statistical section in the methods section, so information is provided only in figure legends.

We apologize for this oversight. Overall, we appreciate the attention to the statistical details from the reviewer in multiple points, which has strengthened the rigor of the manuscript. Now the manuscript includes a “**Statistical analysis**” section in the Methods where we describe in detail the statistical tests that we conducted and how the data is represented in the bar and box plots.

2a. Were the unpaired t-tests one- or two-tailed?

The *t*-tests performed in **Figures 1D, 1F, 3F, and 3G** are unpaired and two-tailed. For clarity and rigor, now we specify the variables of the test in the corresponding figure legends and in the new “**Statistical analysis**” section.

2b. What is meant by “ordinary one-way ANOVA”?

By “ordinary” we meant “regular”, “standard” one-way ANOVA. We apologize for this oversight of letting casual language obscure formal scientific writing. To avoid any misunderstanding with the type of statistical analysis used, we now have removed the adjective “ordinary” from the main text and figure legends.

In Fig 1B, a simple one-way ANOVA at each of three time point can be considered appropriate (although a 2-way ANOVA could have been considered as there are two independent variables (genotype and timepoint) influencing the dependent variable (N:C ratio).

We agree with the reviewer about the possibility to test for one or two variables. In this case we used a one-way ANOVA test because we wanted to determine the differences in N:C ratio between genotypes, independent of developmental time.

However, in Fig 2B, did the “ordinary one-way ANOVA” include Dunnett’s post hoc test (as the p-value bracketing would suggest).

Indeed, we applied a Dunnett’s multiple comparison test to the statistical analysis in **Figures 2B, 4F and 4G**. Now we specify the use of a post hoc test in the figure legend and in the and in the new “**Statistical analysis**” section.

2c. Why show mean \pm SD for Fig 2F when other results are mean \pm SEM?

We thank the reviewer for indicating this oversight in consistency across plots. We have amended the bar plot in **Figure 2F** to display \pm the standard error of the mean like in the other bar plots.

2d. Why are box plots sometimes delineating the 5th and 95th percentiles, and other times the 10th and 90th percentiles?

As in the previous point, we have standardized all the box plots across the manuscript. Now all whiskers delimit 5th to 95th percentiles in the box plots of **Figures 1B, 2B, 3G, and 4F**. This feature of the box plots is described now in the corresponding figure legends and in the new “**Statistical analysis**” section.

2e. It is general unclear how replicated key results are. For example, in the N:C ratios (Fig 1B,2B, 3G, 4F, were all the cells from one single preparation, which would mean that though there are multiple cells, it is only one biological experiment? Were these preparations prepared several times?

Each sample was prepared only once, due to the highly technical and sample-limited process of obtaining blood from millimeter-long embryos. Each blood staining required bleeding ~100 embryos by cutting the tail of the embryo with a sapphire blade. However, biological diversity is ensured by the fact that each N:C ratio analysis comes from the analysis of dozens of cells, that come from the pooled blood of ~100 embryos, which in turn are the mixed offspring of multiple breeding pairs. We have clarified this point in the Methods section.

2f. How many embryos in each group (x with phenotype /y total) demonstrate the WISH phenotypes shown by 2 examples/group in Fig 2D?

We conducted the WISH analysis using 17 wild-type embryos and 17 miR-144 mutant embryos. We did not observe accumulation of *hmgn2* mRNA signal in the posterior blood island in any of the 17 wild-type embryos tested (17/17). We did observe *hmgn2* mRNA signal increase in the posterior blood island in 15 out of 17 miR-144 mutant embryos tested. Since the posterior blood island corresponds to the area of miR-144 expression, these results suggest that the expression of endogenous *hmgn2* is indeed regulated by miR-144. We have now added the number of embryos tested and scored in the main text, the **Figure 2D** and in a **new Figure S2D** panel that shows the whole embryo pictures of the *in situ* hybridization results. It is noted in the figure legend and with asterisks that for two of the embryos shown in the Figure S2B panel, their close-up and cropped tails are shown in Figure 2D.

2g. Again, for Fig 4F, shouldn't this be a one-way ANOVA with Dunnett's post hoc test, rather than an "ordinary one-way ANOVA"?

Indeed, we applied a one-way ANOVA with Dunnett's multiple comparison test to the statistical analysis in Figure 4F. Now we specify the use of a post hoc test in the figure legend and in the new Methods section labeled "**Statistical analysis**".

2h. Fig S1B. The error bars mentioned in the legend are missing from the figure.

Thank you for finding this omission. Now the **Figure S1B** includes the error bars indicating the standard error of the mean.

3. In Figure 3H, I believe that the first and third cell of the five cells arrayed in the TgHmgn2 group is the same cell (and possibly the same image), based on the characteristic indentations in the external cytoplasm, the pattern of punctate lucencies in the nucleus, and a lighter staining cytoplasmic area adjacent to the nucleus from 12 to 3 o'clock.

This observation made me look carefully at the other erythrocyte images, and I also believe that in Fig 2C, for the Gtf2a1 group, the top cell and the second bottom cell are also the same cell (and possibly the same image), based on the eccentric nuclear shape with its broad protrusion at 9 o'clock, a linear lucency in the cytoplasm to the right of the nucleus, and the distribution of thicker darker areas in the cytoplasmic membrane.

While these might individually appear to be minor errors (only 4 examples rather than 5 are provided), the fact that it has occurred twice undermines confidence in all the quantitative N:C

ratio data. Within these datasets, some datapoints of equal value are plotted among the individual values shown in the >90 (or >95) and <10 (or <5) percentile ranges, accepting that the visual resolution in the plot is such that data points might only appear to be equal. The duplicate cells/images within the representative examples suggests that all the N:C image datasets that have been quantified need to be audited for cells photographed twice or for duplicate images.

Also relevant to this point is the question about replication of these experiments in point 2e.

We express our deepest gratitude to the reviewer for catching this unintended mistake with the figure panels so early in the process. We were dismayed to find out the reviewer is correct despite our previous efforts to avoid this type of mistake. As the reviewer noted, our statements were still supported by 4 representative cell images rather than 5 per condition, but nevertheless we have opted to replace those cells with new images in **Figure 2C** and **3H**.

In addition, we have audited the images used for quantifications as requested by the reviewer, and we are now certain that there are no mistakes in the quantifications provided. The audit consisted of the following steps:

- To identify potential duplication of images (i.e., photographed fields)→ Visual inspection and comparison of the cell distribution on the microscopy images reveals no general overlap between any of the images.
- To rule out duplicated cells from partially overlapping fields that might have not been identified in the previous step, we inspected the nucleus and cytoplasm area values for each cell:
 - o If the cells with identical values come from two different fields of view, we eliminated one of the values from the final aggregate table. Since we use the same parameters in ImageJ to quantify each cell across different images, the same cell in two different images would give the same results.
 - o If the cells with identical values come from the same image, we keep both because ultimately, we are measuring very stereotypic variables, and it is reasonable that the values of each condition fluctuate within a narrow range.

After this audition effort, we have found 11 potentially duplicated cells out of 476 total cells in **Figure 2B**, 6 out of 136 in **Figure 3G**, and 9 out of 269 cells in **Figure 4F**. Out of precaution, we have removed these data from our calculations and replotted **Figures 2B**, **3G**, and **4F** and conduct again the corresponding statistical analysis. In addition, the question about replication of these experiments have been addressed in **point 2e**.

Overall, the curated data still fully supports our conclusions.

4. I did not find the schematics helpful, given the compounding negatives. I recognize this might be an individual thing. For example, in Fig3I, in the WT scenario, the negative regulation by miR-144 of Hmgn2 leads to nuclear condensation. However, when miR-144 is removed and Hmgn2 action is increased, the result is LESS Nuclear condensation (although the blue arrow is bigger and it is point to the words “Nuclear condensation”). The schematics are more diagrams of the experimental intervention, rather than schematics indicating the outcome of the experimental intervention – consider, whether the term should be “Decondensation”, or adding an up or down arrow, to help.

We agree with the reviewer that the cartoons as they were drawn in the original submission may lead to some ambiguous interpretations. Therefore, we have amended the cartoons and described in full the interpretation of the cartoons in the figure legend (**Figure 3I** and **4I**) and changed the font to bold to denote increase expression or process.

5. Fig 2B, Dicer1 result. This is a negative result from Dicer1 mRNA injection – it needs to be more firmly technically documented, given the significance made of the finding (p6 lines 17-23). How was it proven that intact Dicer1 mRNA was synthesized and delivered? Were the mRNAs verified as intact at delivery by gel electrophoresis? Was a dose of 100 pg sufficient for this mRNA preparation? One standard way to verify intact RNA delivery is to co-inject all test samples with a fluorophore-encoding mRNA and looking for its expression, and also have a control group of this tracking mRNA only, to control for non-specific RNA effects.

The reviewer raises a fair point. To demonstrate that we overexpressed Dicer activity in the embryos, we have now conducted additional Western blot and Northern blot experiments (**new Figure S2A and S2B**) using the same aliquot of FLAG-tagged Dicer1 mRNA. The Western blot documents the expression of FLAG-tagged Dicer1 in embryos injected with the Dicer1 mRNA. The Northern blot provides a readout of the dicing activity of Dicer1. It demonstrated that when we do a FLAG pull-down from embryos expressing FLAG-Dicer, we retain productive dicing activity. Altogether, these results demonstrate that we overexpressed functional Dicer1 in the embryos and therefore we can claim that overexpression of Dicer1 does not affect the nucleocytoplasmic ratio.

Minor points

1. p15 lines 29-30. Missing references

Thank you for detecting this omission. Now the two references are cited correctly.

2. p16 line 7. Spelling of “glutaraldehyde”

We now have corrected the typo, thank you.

3. P17 line 30. Spelling of “appropriate”

We have corrected the spelling, thank you.

4. I expect the journal will require a scale bar for the erythrocyte pictures, at least initially in Fig 1A and C.

Thank you for pointing us to this requirement. In this new version of the manuscript, we have added the corresponding scale bar to all figure panels with cell imaging (**Figures 1A, 1B, 1C, 2C, 2D, 3H, and 4D**) and the description of the scale bar in the figure legends.

5. Fig 5 legend mentions HBA and HBE, but panels are labelled HBA, HBB and HBG, presumably meaning alpha, beta and gamma.

Figure 5H now shows the qPCR data for the expression of globins HBA, HBB, HBE, and HBE. Now figure and legend are in agreement.

6. The “cDNA preparation and qPCR” section (p21) does not define the primers used for the zebrafish genes.

We attached an accompanying table (**Table S1**) to the new version of the manuscript, where we list the sequences of all oligonucleotides used for cloning or for qPCR. The table is also referenced in the “cDNA preparation and qPCR” section.

REVIEWERS' COMMENTS

Reviewer #1 (Remarks to the Author):

The authors have responded to my comments suggestion and I'm happy to support publication.

Reviewer #2 (Remarks to the Author):

The m/s is much improved.

RESPONSES

1. Mammalian data

Experiments have been added that strengthen the claim of conservation in mammalian systems (Fig 5 and related supplementary data).

2. N:C ratio images and data audit

The authors have conducted a self-audit, detailed in their response, which appears to acknowledge that fields scored were sometimes overlapping. This and other potential duplications have been rectified.

3. Negative result from Dicer overexpression.

Additional data are provided to verify that the experimental system could have detected a dicer-dependent effect.

4. Statistics

The statistical presentation is much more consistent.

MINOR ADDITIONAL POINTS

1. Spelling

Fig 3A – brightfield is mis-spelt twice

Fig 2G – erythrocytes is mis-spelt multiple times

Fig 5D – transgene is mis-spelt

2. Fig2D – scale bar. This scale bar is unlikely to represent 1 mm, as it would mean the whole embryo is very much longer than 3mm; Kimmel et al 1995 gives 2.5 mm as the tight mean length of a 30 hpf embryo. https://zfin.org/zf_info/zfbook/stages/figs/fig16.html

3. Fig 3C – a minor detail not affecting function, but I believe a single LoxP site will remain between EYFP and SV40 polyA in the recombined allele (it is absent in the figure)

4. The method of o-dianisidine quantification described is area-based (p28 lines 22-27). This is fine as an

approach, and the axis labels are OK in the figure itself, but should not be referred to as “intensity” in the legend (Fig 4 legend, line 16)

RESPONSE TO THE REVIEWERS

We want to take this opportunity to collectively thank the reviewers for their comments, which have guided us to further improve the rigor and clarity of the manuscript.

The original comments of the reviewers are listed in **black**; our responses are written in **blue**.

Reviewer #1 (Remarks to the Author):

The authors have responded to my comments suggestion and I'm happy to support publication.

We are glad to hear that we responded to the full satisfaction of the reviewer and thank once more their contribution to the improvement of the manuscript.

Reviewer #2 (Remarks to the Author):

The m/s is much improved.

We sincerely appreciate that the reviewer recognizes our effort in improving the manuscript and we want to tank for helping us with their comments.

RESPONSES

1. Mammalian data

Experiments have been added that strengthen the claim of conservation in mammalian systems (Fig 5 and related supplementary data).

2. N:C ratio images and data audit

The authors have conducted a self-audit, detailed in their response, which appears to acknowledge that fields scored were sometimes overlapping. This and other potential duplications have been rectified.

3. Negative result from Dicer overexpression.

Additional data are provided to verify that the experimental system could have detected a dicer-dependent effect.

4. Statistics

The statistical presentation is much more consistent.

MINOR ADDITIONAL POINTS

1. Spelling

Fig 3A – brightfield is mis-spelt twice.

We thank the reviewer for noticing this typo. We corrected the spelling of “brightfield” in a new version of the figure.

Fig 2G – erythrocytes is mis-spelt multiple times

We thank the reviewer for noticing this typo. We corrected the spelling of “erythrocytes” in a new version of the figure.

Fig 5D – transgene is mis-spelt

We thank the reviewer for noticing this typo. We corrected the spelling of “transgene” in a new version of the figure.

2. Fig2D – scale bar. This scale bar is unlikely to represent 1 mm, as it would mean the whole embryo is very much longer than 3mm; Kimmel et al 1995 gives 2.5 mm as the tight mean length of a 30 hpf embryo. https://zfin.org/zf_info/zfbook/stages/figs/fig16.html

We appreciate that the reviewer pointed to this oversight, and we amended the information about the scale bar. Now it says 0.2 mm instead of 1 mm in the figure legend.

3. Fig 3C – a minor detail not affecting function, but I believe a single LoxP site will remain between EYFP and SV40 polyA in the recombined allele (it is absent in the figure)

The reviewer is correct. We modified the figure 3C to indicate the remaining LoxP site.

4. The method of o-dianisidine quantification described is area-based (p28 lines 22-27). This is fine as an approach, and the axis labels are OK in the figure itself, but should not be referred to as “intensity” in the legend (Fig 4 legend, line 16)

The Y axes has been modified and say “Relative quantification of O-dianisidine staining”.